# Vglut2-based glutamatergic signaling in central noradrenergic neurons is dispensable for normal breathing and chemosensory reflexes

Yuan Chang[1,2], Savannah Lusk[1], Andersen Chang[1], Christopher S Ward[2], Russell S Ray[1,2,3]*

[1]Department of Neuroscience, Baylor College of Medicine, Houston, United States; [2]Department of Integrative Physiology, Baylor College of Medicine, Houston, United States; [3]McNair Medical Institute, Houston, United States

*For correspondence:
Russell.Ray@bcm.edu

Competing interest: The authors declare that no competing interests exist.

**Abstract** Central noradrenergic (NA) neurons are key constituents of the respiratory homeostatic network. NA dysfunction is implicated in several developmental respiratory disorders including Congenital Central Hyperventilation Syndrome (CCHS), Sudden Infant Death Syndrome (SIDS), and Rett Syndrome. The current unchallenged paradigm in the field, supported by multiple studies, is that glutamate co-transmission in subsets of central NA neurons plays a role in breathing control. If true, NA-glutamate co-transmission may also be mechanistically important in respiratory disorders. However, the requirement of NA-derived glutamate in breathing has not been directly tested and the extent of glutamate co-transmission in the central NA system remains uncharacterized. Therefore, we fully characterized the cumulative fate maps and acute adult expression patterns of all three vesicular glutamate transporters (*Slc17a7* (Vglut1), *Slc17a6* (Vglut2), and *Slc17a8* (Vglut3)) in NA neurons, identifying a novel, dynamic expression pattern for Vglut2 and an undescribed co-expression domain for Vglut3 in the NA system. In contrast to our initial hypothesis that NA-derived glutamate is required to breathing, our functional studies showed that loss of Vglut2 throughout the NA system failed to alter breathing or metabolism under room air, hypercapnia, or hypoxia in unrestrained and unanesthetized mice. These data demonstrate that Vglut2-based glutamatergic signaling within the central NA system is not required for normal baseline breathing and hypercapnic, hypoxic chemosensory reflexes. These outcomes challenge the current understanding of central NA neurons in the control of breathing and suggests that glutamate may not be a critical target to understand NA neuron dysfunction in respiratory diseases.

## eLife assessment

Chang et al. provide glutamate co-expression profiles in the central noradrenergic system and test the requirement of Vglut2-based glutamatergic release in respiratory and metabolic activity under physiologically relevant gas challenges. Their experiments provide **compelling** evidence that conditional deletion of vesicular glutamate transporters from noradrenergic neurons does not impact steady-state breathing or metabolic activity in room air, hypercapnia, or hypoxia. This study provides an **important** contribution to our understanding of how noradrenergic neurons regulate respiratory homeostasis in conscious adult mice.

## Introduction

Breathing is a vital and life-sustaining function supporting homeostatic processes, most critically maintaining blood $pH/CO_2$ and $O_2$ levels within a narrow physiological range. Respiratory homeostasis is mediated by neuron-modulated lung ventilation adjustments in response to physiological deviations resulting in high $pCO_2$ or low $pO_2$ blood and tissue levels, known as the hypercapnic and hypoxic reflexes, respectively (*Del Negro et al., 2018*; *Dick et al., 2018*). These reflexes are part of a complex brainstem neural network that integrates a multitude of information streams across the central and peripheral nervous systems to regulate respiratory output. Within this brainstem network, central noradrenergic (NA) neurons are known to be an important component that plays a variety of roles in modulating breathing. Furthermore, various perturbations across the central NA system have been implicated in several developmental disorders with respiratory and chemosensory features such as Congenital Central Hyperventilation Syndrome (CCHS), Sudden Infant Death Syndrome (SIDS), and Rett Syndrome (*Beltrán-Castillo et al., 2017*; *Feldman et al., 2013*; *Gauda et al., 2007*; *Viemari, 2008*). Thus, understanding how central NA neurons modulate respiratory chemoreflexes or chemosensory breathing is critically important for the development of new diagnostic and therapeutic interventions to address respiratory pathophysiology.

Central NA neurons are commonly thought to exert their effect on breathing through their primary neurotransmitter noradrenaline and adrenaline. However, in addition to noradrenaline, several studies provide strong evidence that glutamate is co-transmitted in subsets of central NA neurons. Vesicular glutamate transporter 2 (Vglut2), a gene marker of glutamatergic signaling, has been shown to be co-expressed in subsets of central NA neurons including, C1/A1, C2/A2, A5, and LC in adult rats and mice (*DePuy et al., 2013*; *Souza et al., 2022a*; *Stornetta et al., 2002a*; *Stornetta et al., 2002b*; *Yang et al., 2021*), though NA-specific expression of related Vglut1 and Vglut3 transporters remains unknown. Additionally, it has been well documented that central NA neurons co-expressing Vglut2 innervate key respiratory centers, such as preBötzinger complex and parafacial region (pFRG). Additionally, central NA neurons co-expressing Vglut2 project to many key autonomic brainstem, spinal cord, and forebrain centers, such as the dorsal motor nucleus of the vagus, intermediolateral nucleus and sympathetic system, and the hypothalamus, all of which could drive a change in breathing if perturbed, that is, indirect effects from cardiovascular and metabolic dysregulation (*Supplementary file 1*). Thus, it has become a dominant paradigm in the field of respiratory physiology that NA-based glutamate release is a critical form of neurotransmission in the control of breathing. *Abbott et al., 2014* showed that Vglut2 is required for an increase in respiratory rate when anterior C1 neurons are unilaterally optogenetically stimulated. *Malheiros-Lima et al., 2020* showed that Vglut2-expressing C1 neurons project to the pFRG region, and hypoxic breathing was blunted after blockade of ionotropic glutamatergic receptors at the pFRG site in anesthetized rats, together supporting a role for anterior C1 neurons releasing glutamate at the pFRG site to regulate breathing under hypoxia. Similarly, *Malheiros-Lima et al., 2022* and *Malheiros-Lima et al., 2018* showed that Vglut2-expressing C1 neurons project to the NA A5 region and the preBötzinger complex, and, again, the blockade of ionotropic glutamatergic receptors at the A5 region or preBötzinger complex reduced the increase in phrenic nerve activity and respiratory frequency elicited by optogenetic stimulation of C1 cells in an anesthetized preparation. In addition, *Guyenet et al., 2013* and others speculated that the apparent lack of plasmalemmal monoamine transporter in C1 fibers indicates reduced or absent NA or adrenergic signaling due a lack of re-uptake and neurotransmitter pool depletion (*Comer et al., 1998*; *Lorang et al., 1994*). Cumulatively, the studies argue for glutamate as the predominant functional neurotransmitter for C1 NA neurons in the breathing neural network. To our knowledge, this dominant perspective has not been otherwise previously challenged. Although these studies are informative, the evidence supporting the role of NA-based Vglut2 signaling in respiratory control are either indirect and circumstantial or cannot be seen as physiological given experimental limitations, such as the focal nature of optogenetic stimulation. Thus, it is not yet clear what the requirement is for NA-based glutamatergic signaling in homeostatic breathing in the unanesthetized and unrestrained animal and how that might inform upon disease.

To better understand the role of NA-based glutamatergic signaling in breathing, we sought to both fully characterize the molecular profiles of the central NA system with respect to glutamate co-expression and to test the hypothesis that Vglut2-based glutamatergic release is required in respiratory control under physiological chemosensory challenges in unanesthetized and unrestrained mice. We

first fully characterized the recombinase-based cumulative fate maps for Vglut1, Vglut2, and Vglut3 expression and compared those maps to their real-time expression profiles in central NA neurons by RNA in situ hybridization in adult mice. We found a novel dynamic expression pattern for Vglut2 and an entirely undescribed co-expression domain for Vglut3 in the central NA system. Second, to determine if Vglut2-based glutamatergic signaling in NA neurons is required for respiratory homeostasis, we conditionally ablated Vglut2 in all NA neurons and tested respiratory, chemosensory, and metabolic function in unrestrained and unanesthetized mice. Using the same genetic model in prior studies (*Abbott et al., 2014*), conditional deletion of Vglut2 in NA neurons did not significantly impact breathing under room air, hypercapnic, or hypoxic conditions. These results demonstrate, for the first time, that NA Vglut2-based glutamatergic signaling is dispensable for respiratory control, which challenges the prevalent perspective on the role of C1 and other Vglut2-expressing NA neurons in respiratory homeostasis and suggests that glutamate may not be a critical target to understand NA neuron dysfunction in respiratory diseases.

## Results
### Cumulative fate maps of central NA neurons co-expressing Vglut1, Vglut2 and Vglut3

To fully characterize the expression profiles of all three glutamate markers Vglut1, Vglut2, and Vglut3 in central NA neurons, we first used an intersectional genetic strategy (*Figure 1A*). We bred three Cre drivers *Slc17a7$^{Cre}$* (Vglut1-Cre), *Slc17a6$^{Cre}$* (Vglut2-Cre), and *Slc17a8$^{Cre}$* (Vglut3-Cre) to *Dbh$^{p2a-Flpo}$* (DBH-p2a-Flpo) (targeting NA neurons) mice, respectively. The three compound lines of *Slc17a7$^{Cre}$*; *Dbh$^{p2a-Flpo}$*, *Slc17a6$^{Cre}$*; *Dbh$^{p2a-Flpo}$*, and *Slc17a8$^{Cre}$*; *Dbh$^{p2a-Flpo}$* were then crossed with the *Rosa26$^{RC::FLTG}$* (RC::FLTG). RC::FLTG are intersectional reporter mice that express tdTomato in cells expressing only flippase (Flpo) and express eGFP in cells co-expressing both Flpo and Cre recombinases. Thus, in each of the three intersectional reporter crosses [e.g. (1) *Slc17a7$^{Cre}$*; *Dbh$^{p2a-Flpo}$*; *Rosa26$^{RC::FLTG}$*, (2) *Slc17a6$^{Cre}$*; *Dbh$^{p2a-Flpo}$*; *Rosa26$^{RC::FLTG}$*, and (3) *Slc17a8$^{Cre}$*; *Dbh$^{p2a-Flpo}$*; *Rosa26$^{RC::FLTG}$*], NA neurons without any Vglut1, 2, or 3 expression are labeled by red fluorescent protein (tdTomato) while NA neurons co-expressing either Vglut1, Vglut2, or Vglut3 are labeled by green fluorescent protein (eGFP). By using this method, we characterized and quantified the expression profiles of all three vesicular glutamate transporters Vglut1, Vglut2, and Vglut3 across every anatomically defined NA nucleus in adult mice including A7, Locus Coeruleus (LC), dorsal/ventral subcoeruleus nucleus (sub CD/CV), A5, and the anterior–posterior dimensions of C1/A1, C2/A2 (*Robertson et al., 2013*), and C3 (*Figure 1B–D* and *Figure 1— figure supplement 1A, B*). Vglut1-Cre was not co-expressed in any central NA nuclei in adult mice. However, Vglut2-Cre and Vglut3-Cre both showed co-expression in central NA neurons. Vglut3-Cre-expressing NA neurons were restricted to posterior C2/A2 and posterior C1/A1, with greatest expression in posterior C2/A2 where 26.9 ± 3.16% of NA neurons were Vglut3-Cre positive. In posterior C1/A1, only 1.26 ± 0.559% of NA neurons were Vglut3-Cre positive. Surprisingly, 84.6 ± 3.75% of NA neurons in total showed Vglut2-Cre co-expression and each NA nucleus was predominantly labeled by Vglut2-Cre expression. Over 50% of NA neurons in each NA nucleus were Vglut2-Cre positive. The percentages of Vglut2-Cre positive NA neurons in each NA nucleus were as follows: 69.8 ± 8.29% in A7, 80.5 ± 3.78% in LC, 84.8 ± 11.3% in A5, 51.5 ± 2.61% in sub CD/CV, 99.0 ± 0.681% in anterior C1/A1, 95.7 ± 2.40% in posterior C1/A1, 96.1 ± 1.98% in anterior C2/A2, 97.8 ± 0.771% in posterior C2/A2, and 100 ± 0.00% in C3 (mean ± the standard error of the mean (SEM)). The presence of Vglut2-Cre co-expression in anterior NA groups was unexpected as previous in situ data in adult rats found *Slc17a6* (Vglut2) positive NA neurons only in the posterior C2/A2 and C1/A1 (*Stornetta et al., 2002a*; *Stornetta et al., 2002b*). Notably, however, the Vglut2 co-expression in the LC region agrees with *Yang et al., 2021* which also showed that 89% of LC NA neurons are Vglut2 positive using a different intersectional strategy in mice: *Th$^{Flpo}$*; *Slc17a6$^{Cre}$*; *Rosa26$^{Ai65}$* (TH-Flpo; Vglut2-Cre; Ai65). The differences in our and other fate maps compared to in situ hybridization may either reflect early gene expression that is downregulated in the adult or may reflect low levels of expression not detectable by in situ hybridization but that are nonetheless sufficient to affect recombination in the intersectional genetic strategy.

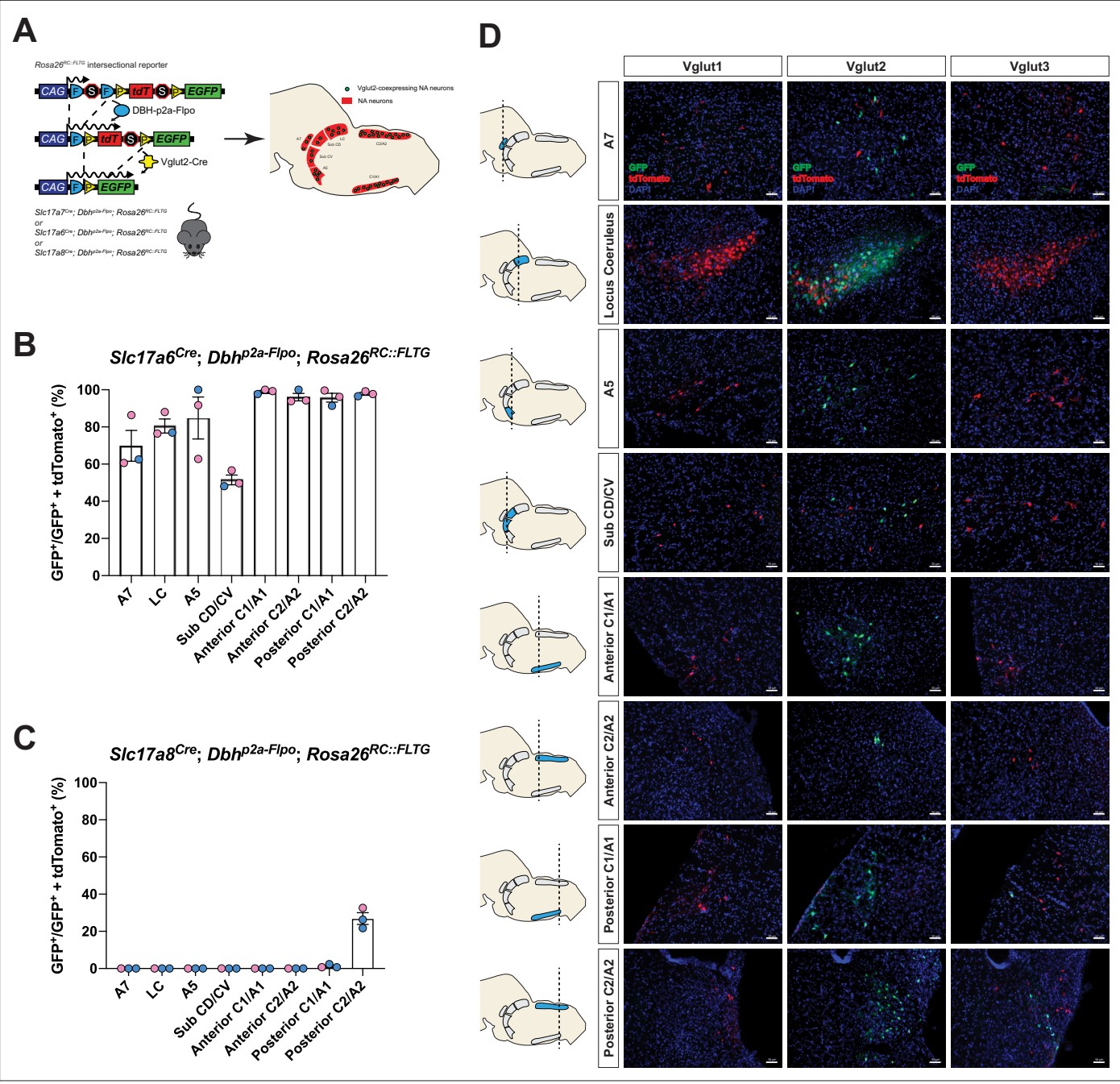

**Figure 1.** Cumulative fate maps of glutamate co-expressing noradrenergic (NA) neurons characterized by intersectional genetics. (**A**) Breeding schematic to generate the three intersectional reporter lines $Slc17a7^{Cre}$; $Dbh^{p2a-Flpo}$; $Rosa26^{RC::FLTG}$, $Slc17a6^{Cre}$; $Dbh^{p2a-Flpo}$; $Rosa26^{RC::FLTG}$, $Slc17a8^{Cre}$; $Dbh^{p2a-Flpo}$; $Rosa26^{RC::FLTG}$. In each of the three intersectional reporter crosses, NA neurons co-expressing either Vglut1, Vglut2, or Vglut3 are labeled by green fluorescent protein (eGFP) while NA neurons without any Vglut1, 2, or 3 expressions are labeled by red fluorescent protein (tdTomato). Shown as an example is the $Slc17a6^{Cre}$; $Dbh^{p2a-Flpo}$; $Rosa26^{RC::FLTG}$ (Vglut2-Cre/+; DBH-p2a-Flpo/+; RC::FLTG/+) intersectional reporter line. (**B**) Quantification of the percentage of Vglut2-coexpressing NA neurons among NA neurons in each brainstem NA nuclei including A7, LC, A5, sub CD/CV, anterior C1/A1 and C2/A2, posterior C1/A1 and C2/A2. Pink data points represent female data while blue data points represent male data. (**C**) Quantification of the percentage of Vglut3-coexpressing NA neurons among NA neurons in each NA nucleus. Pink data points represent female data while blue data points represent male data. (**D**) Fluorescent expression of tdTomato (red) and eGFP (green) in coronal sections of Vglut1, Vglut2, or Vglut3 intersectional reporter lines in brainstem NA nuclei. Nucleus is labeled by 4',6-diamidino-2-phenylindole (DAPI) (blue). Scale bar 50 μm.

The online version of this article includes the following source data and figure supplement(s) for figure 1:

**Source data 1.** Cell quantification for Vglut2-coexpressing noradrenergic (NA) neurons in the cumulative fate map.

*Figure 1 continued on next page*

*Figure 1 continued*

**Source data 2.** Cell quantification for Vglut3-coexpressing noradrenergic (NA) neurons in the cumulative fate map.

**Figure supplement 1.** Cumulative fate maps of glutamate co-expressing noradrenergic (NA) neurons in C3 NA nucleus.

## Real-time mRNA expression patterns of Vglut1, Vglut2, and Vglut3 in central NA neurons in adult mice

To verify if the real-time NA-based Vglut2 expression pattern in adult mice is comparable to previous reports in adult rats (*DePuy et al., 2013*; *Stornetta et al., 2002a*; *Stornetta et al., 2002b*), and to further characterize the expression patterns of the other two vesicular glutamate transporters (Vglut1 and Vglut3) in adult mice, we performed fluorescent RNA in situ hybridization experiments. We co-stained *Slc17a7* (Vglut1), *Slc17a6* (Vglut2), or *Slc17a8* (Vglut3) with *Dbh*, respectively, in brain tissue of adult mice and characterized the colocalization of *Slc17a7*, *Slc17a6*, *Slc17a8* with *Dbh* in brainstem NA nuclei A7, LC, A5, sub CD/CV, anterior C1/A1 and C2/A2, posterior C1/A1 and C2/A2, and C3 (*Figure 2A–C* and *Figure 2—figure supplement 2A, B*). No *Slc17a7/Dbh* double positive neurons were detected in any part of the central NA system, consistent with our fate map. *Slc17a8* mRNA colocalization with *Dbh* was only found in the posterior part of C2/A2 where 27.1 ± 1.86% of *Dbh* positive neurons demonstrated *Slc17a8* colocalization, again consistent with our fate map. However, we only found detectable levels of *Slc17a6* mRNA in the C1/A1, C2/A2, and C3 NA regions, but not in the A7, LC, A5, sub CD/CV. The percentages of *Slc17a6/Dbh* double positive neurons for regions where detected were as follows: 84.7 ± 5.82% in anterior C1/A1, 66.3 ± 0.335% in anterior C2/A2, 35.7 ± 3.01% in posterior C1/A1, 90.1 ± 2.45% in posterior C2/A2, and 79.7 ± 3.94% in C3. This result is consistent with the previous in situ data in adult rats (*Stornetta et al., 2002a*; *Stornetta et al., 2002b*) suggesting that there is no obvious expression difference of *Slc17a6* in the central NA system between mouse and rat. However, the *Slc17a6* in situ data did not show expression in anterior NA populations. This difference between our in situ data and fate map data supports our previous hypothesis that many NA neurons expressed *Slc17a6* (Vglut2) at some point from early development toward adulthood but the expression of *Slc17a6* (Vglut2) is diminished during adulthood. Interestingly, *Yang et al., 2021* did show adult Vglut2 co-expression in LC by a viral injection of both Cre and Flpo-dependent eYFP into the LC of a bi-transgenic mouse with both *Th^Flpo^* and *Slc17a6^Cre^*. To verify this result, we injected an AAV virus containing Cre-dependent tdTomato (pAAV-EF1a-DIO-tdTomato-WPRE) into the LC region of adult *Slc17a6^Cre^* mice. Consistent with *Yang et al., 2021*, we observed sparse Vglut2-Cre and TH-immunopositive double labeled neurons in the LC (*Figure 2—figure supplement 1*). Additionally, *Souza et al., 2022b* showed about 28.5% of A5 neurons are *Slc17a6* mRNA positive in adult rats by RNA scope, a method which is more sensitive than the traditional in situ hybridization we used here. These data suggest that some adult NA neurons in LC and A5 have *Slc17a6* (Vglut2) co-expression, but the *Slc17a6* mRNA level cannot be detected by traditional in situ hybridization.

## Vglut2 expression is effectively knocked down in the whole central NA system in Dbh-Cre; Vglut2 cKO mice

To investigate the requirement of Vglut2-based glutamatergic signaling in the central NA system in breathing under physiological challenges, we used Dbh-Cre; *Slc17a6^flox/flox^* (Dbh-Cre; Vglut2 cKO) to remove Vglut2 expression from all NA neurons. This is the same model used by *Abbott et al., 2014* to show that Vglut2 expression in the NA system was required for an increase in respiratory frequency following unilateral optogenetic stimulation of anterior C1. To verify if Vglut2 expression was effectively removed from central NA neurons in Dbh-Cre; Vglut2 cKO mice, we used fluorescent mRNA in situ hybridization for *Slc17a6* (Vglut2) and *Dbh* in the mouse brainstems and compared the *Slc17a6* (Vglut2) signal intensity in *Dbh* positive neurons in Dbh-Cre; Vglut2 cKO mice to their littermate controls. We found that *Slc17a6* mRNA expression was 94.7 ± 3.97% decreased compared to controls, indicating that the *Slc17a6* (Vglut2) was effectively recombined to abrogate expression in NA neurons (*Figure 3A, B* and *Figure 3—figure supplement 1*).

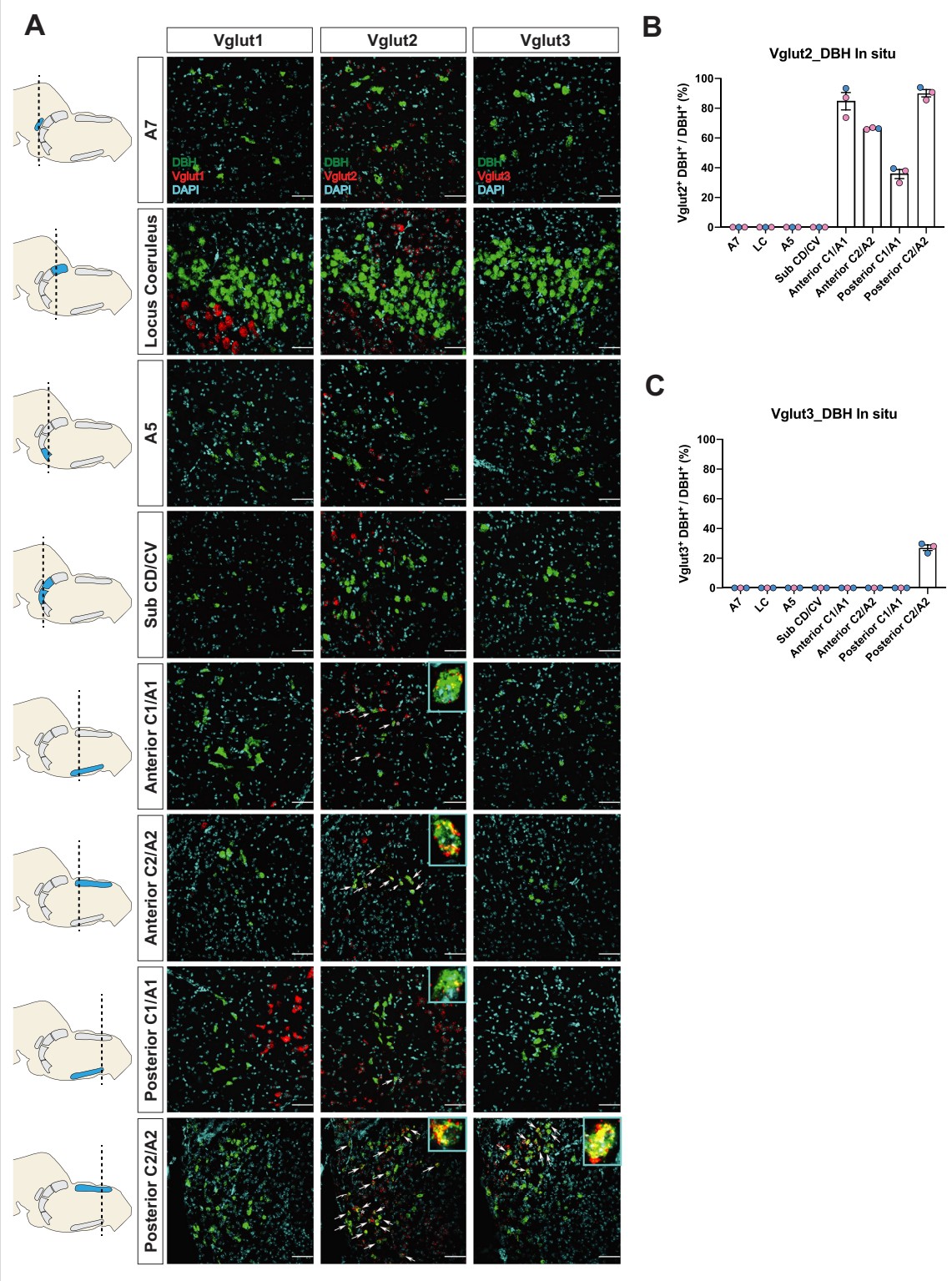

**Figure 2.** Characterization of real-time co-expression of Vglut1, Vglut2, and Vglut3 in brainstem noradrenergic (NA) neurons in adult mice by fluorescent in situ hybridization. (**A**) Representative images of *Slc17a7* (Vglut1), *Slc17a6* (Vglut2), and *Slc17a8* (Vglut3) and *Dbh* (DBH) double ISH in coronal sections of WT mice in all of brainstem NA nuclei in adult mice. DBH (green), Vglut1/2/3 (red), DAPI (cyan). Scale bar 50 µm. (**B**) Quantification of the percentage of *Slc17a6* (Vglut2) co-expression in NA neurons in each NA nucleus. Female data (pink), male data (blue). (**C**) Quantification of the percentage of *Slc17a8* (Vglut3) co-expression in NA neurons in each NA nucleus. Female data (pink), male data (blue).

The online version of this article includes the following source data and figure supplement(s) for figure 2:

*Figure 2 continued on next page*

*Figure 2 continued*

**Source data 1.** Cell quantification for Vglut2-coexpressing noradrenergic (NA) neurons in the in situ hybridization.

**Source data 2.** Cell quantification for Vglut3-coexpressing noradrenergic (NA) neurons in the in situ hybridization.

**Figure supplement 1.** Virus injection of Cre-dependent tdTomato into LC of *Slc17a6*[Cre] mice.

**Figure supplement 2.** Characterization of real-time co-expression of Vglut1, Vglut2, and Vglut3 in C3 noradrenergic (NA) neurons in adult mice by fluorescent in situ hybridization.

## Vglut2-based glutamatergic signaling in central NA neurons is not required for baseline breathing nor for the hypercapnic ventilatory reflex under distinct CO₂ challenges

Multiple prior studies provide indirect evidence that NA glutamate transmission may play roles in respiratory function, particularly that anterior C1 neurons mediate the hypoxic ventilatory response through the pFRG/RTN and/or the NA A5 group (*Malheiros-Lima et al., 2020*; *Malheiros-Lima et al., 2022*). Additionally, the A5 group has been implicated in the hypercapnic reflex (*Haxhiu et al., 1996*). Thus, we sought to determine if Vglut2-based glutamatergic signaling in central NA neurons

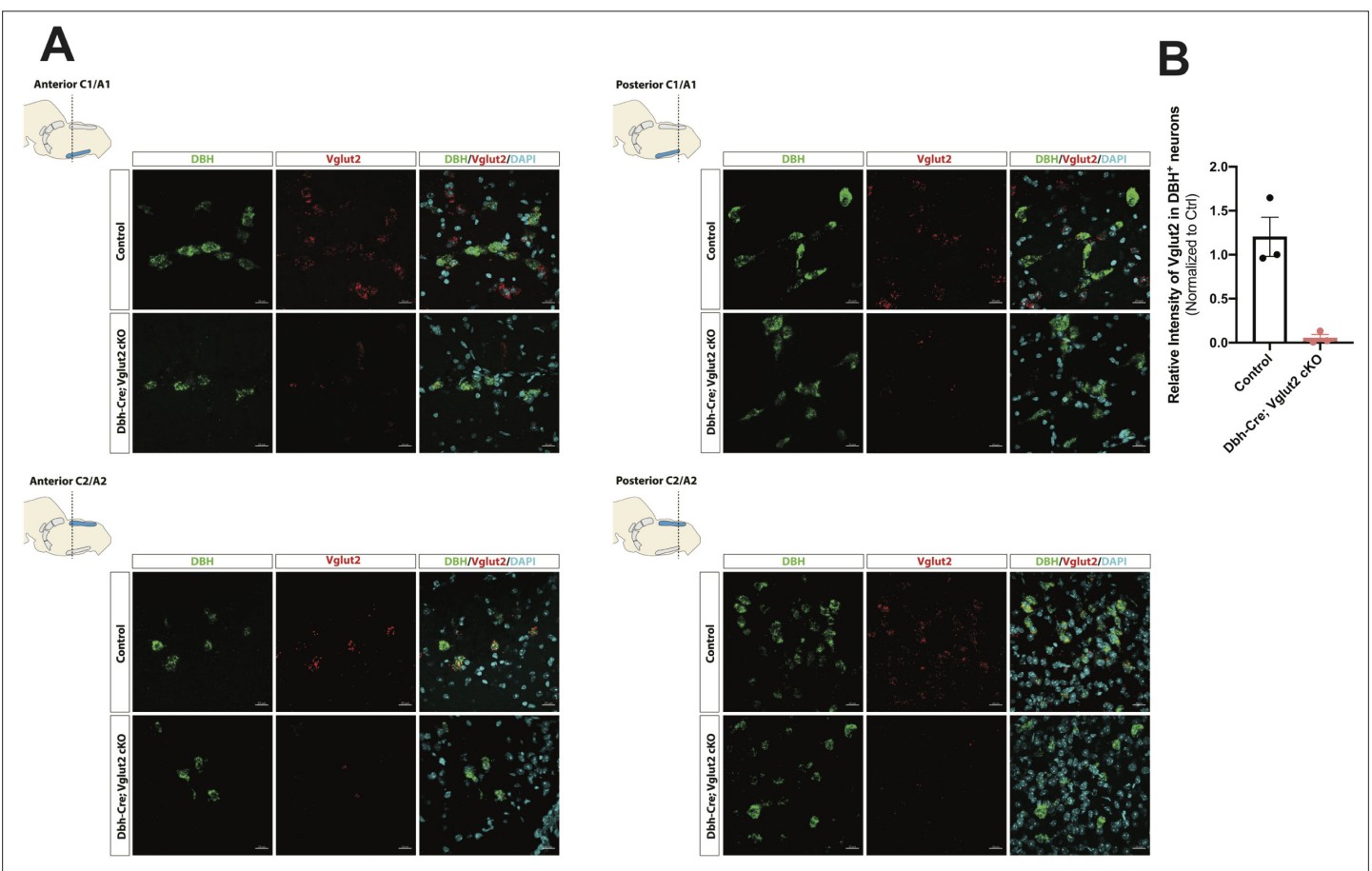

**Figure 3.** DBH with Vglut2 co-staining by fluorescent in situ hybridization confirms that Vglut2 expression is disrupted in the whole central noradrenergic (NA) system in Dbh-Cre; Vglut2 cKO mice. (**A**) Representative images of *Slc17a6* (Vglut2) and *Dbh* (DBH) double ISH in anterior C1/A1, posterior C1/A1, anterior C2/A2, and posterior C2/A2 in control and Dbh-Cre; Vglut2 cKO mice. DBH (green), Vglut2 (red), DAPI (cyan). Scale bar 20 μm.(**B**) Quantification of intensity of *Slc17a6* (Vglut2) signal in control and Dbh-Cre; Vglut2 cKO mice.

The online version of this article includes the following source data and figure supplement(s) for figure 3:

**Source data 1.** Image quantification for Vglut2 signal intensity in Dbh-Cre; Vglut2 cKO and their littermate controls.

**Figure supplement 1.** DBH with Vglut2 co-staining by fluorescent in situ hybridization confirms that Vglut2 expression is disrupted in C3 noradrenergic (NA) nucleus in Dbh-Cre; Vglut2 cKO mice.

is necessary to regulate baseline breathing and the hypercapnic chemoreflex in unanesthetized and unrestrained animals. We used whole-body barometric plethysmography to measure the breathing and metabolism of unanesthetized and unrestrained mice. After a 5-day habituation protocol, the respiratory responses of Dbh-Cre; Vglut2 cKO mice and their littermate controls, including the overall respiratory output or ventilatory equivalent of oxygen ($V_E/V_{O_2}$), respiratory rate ($V_f$), tidal volume ($V_T$), minute ventilation ($V_E$), metabolic demand ($V_{O_2}$), inspiratory duration ($T_I$), expiratory duration ($T_E$), breath cycle duration ($T_{TOT}$), inspiratory flow rate ($V_T/T_I$), and expiratory flow rate ($V_T/T_E$), were measured under room air and hypercapnia (5% $CO_2$) (**Figure 4A**). 5% $CO_2$ gas challenge produced a significant change in all the respiratory parameters compared to room air in both Dbh-Cre; Vglut2 cKO and control mice. However, surprisingly, Dbh-Cre; Vglut2 cKO mice did not show significant differences in any respiratory or metabolic parameter mentioned above compared to their littermate controls under both room air and 5% $CO_2$ challenges (**Figure 4B** and **Tables 1 and 2**). In addition, to investigate if Vglut2-based NA-derived glutamate is required for regulating the dynamic patterns of breathing, such as the breathing rhythms, we generated Poincaré plots for $T_I$, $T_E$, $T_{TOT}$, $V_T/T_I$, $V_T/T_E$, $V_f$, $V_T$, and $V_E$ and we calculated the SD1 and SD2 statistics for each parameter under room air and 5% $CO_2$. We found that Dbh-Cre; Vglut2 cKO mice only showed a marginally statistically significant increase in the SD1 of inspiratory duration ($T_I$) (p = 0.0492). Outside of this single comparison, however, we found no significant differences in the other parameters between the two groups (**Figure 4—figure supplement 1A–H**). Thus, we theorize that the statistically significant result for the SD1 of $T_I$ seen here is likely either a result of measurement noise or a consequence of a type 1 error. The overall pattern of results suggested that Vglut2-based glutamatergic signaling is not required for regulating either the steady state of or the dynamic patterns of baseline breathing and hypercapnic ventilatory reflex under 5% $CO_2$.

To further investigate the role of Vglut2-based glutamatergic signaling in the hypercapnic response, we interrogated the system further with more severe hypercapnic exposures and measured the breathing of Dbh-Cre; Vglut2 cKO mice under 7% and 10% $CO_2$. Both 7% and 10% $CO_2$ stimuli produced a significant respiratory response in both mutant and control mice. Under the 7% $CO_2$ condition, Dbh-Cre; Vglut2 cKO mice showed a significantly decreased tidal volume ($V_T$) (p = 0.023) and a significantly reduced inspiratory flow rate ($V_T/T_I$) (p = 0.0016), but the overall respiratory output ($V_E/V_{O_2}$), respiratory rate ($V_f$), minute ventilation ($V_E$), metabolic demand ($V_{O_2}$), inspiratory duration ($T_I$), expiratory duration ($T_E$), breath cycle duration ($T_{TOT}$), and expiratory flow rate ($V_T/T_E$) did not show a significant difference between the mutant mice and their sibling controls (**Figure 5A, B** and **Table 3**). Additionally, none of the dynamic patterns of any respiratory parameter showed a significant difference between the mutant and control groups (**Figure 5—figure supplement 1A–H**). This result suggests that Vglut2-based glutamatergic signaling in central NA neurons is not required for regulating the hypercapnic ventilatory reflex and breathing regularity under 7% $CO_2$.

Under 10% $CO_2$ challenge, none of the steady-state breathing parameters including the overall respiratory outputs ($V_E/V_{O_2}$), respiratory rate ($V_f$), tidal volume ($V_T$), minute ventilation ($V_E$), metabolic demand ($V_{O_2}$), inspiratory duration ($T_I$), expiratory duration ($T_E$), breath cycle duration ($T_{TOT}$), inspiratory flow rate ($V_T/T_I$), and expiratory flow rate ($V_T/T_E$) were significantly different between Dbh-Cre; Vglut2 cKO mice and their littermate controls, suggesting that Vglut2-based glutamatergic signaling in central NA neurons is not required for the hypercapnic ventilatory reflex under 10% $CO_2$ (**Figure 6A, B** and **Table 4**). For the dynamic patterns of breathing, the SD1 of breath cycle duration ($T_{TOT}$) (p = 0.028) and SD1 of respiratory rate ($V_f$) (p = 0.038) showed a significant increase in mutant mice compared to the controls (**Figure 6—figure supplement 1A–H**). This suggests that removing Vglut2 from central NA neurons may increase the breathing irregularity under high $CO_2$ challenges like 10% $CO_2$.

## Vglut2-based glutamatergic signaling in central NA neurons is not required for the hypoxic ventilatory reflex under 10% $O_2$

To determine if Vglut2-based glutamatergic signaling in central NA neurons is necessary to regulate the hypoxic chemoreflex, we measured the ventilation response of Dbh-Cre; Vglut2 cKO mice under 10% $O_2$. We analyzed the ten respiratory parameters in three 5 min epochs following 10% $O_2$ challenge onset. Similar to the hypercapnic challenges, the 10% $O_2$ challenge produced a significant breathing response in every 5 min epoch compared to room air, but no significant difference was found in the overall respiratory output ($V_E/V_{O_2}$), respiratory rate ($V_f$), tidal volume ($V_T$), minute ventilation ($V_E$),

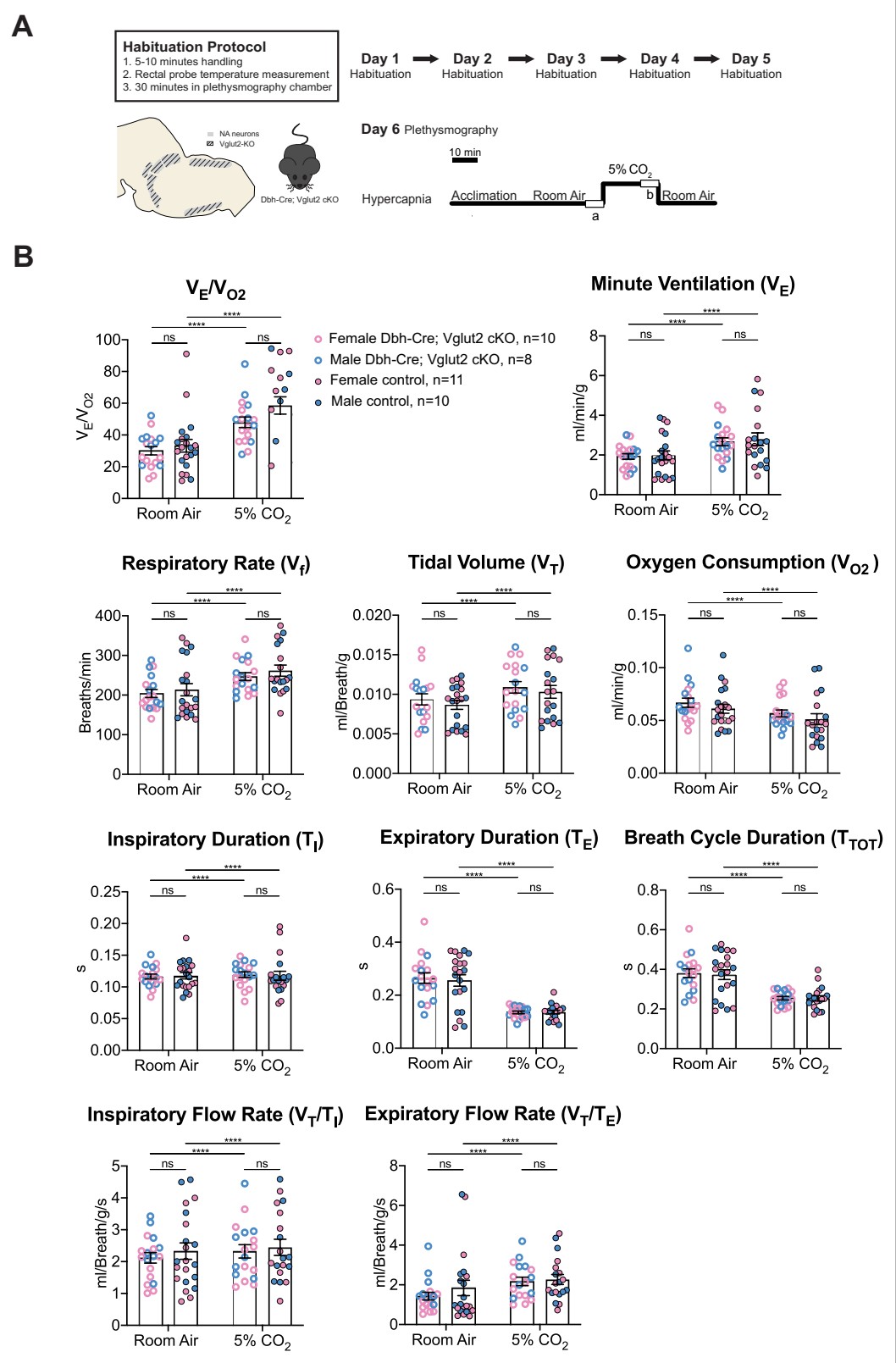

**Figure 4.** Vglut2 conditional knockout in central noradrenergic neurons fails to alter baseline breathing and the hypercapnic ventilatory reflex (5% $CO_2$). (**A**) Mouse model schematic and experimental protocol including habituation and hypercapnia protocol (5% $CO_2$). (**B**) Under both room air and hypercapnia (5% $CO_2$), Dbh-Cre; Vglut2 cKO mice did not show significant changes in respiratory output ($V_E/V_{O_2}$), minute ventilation ($V_E$), respiratory

*Figure 4 continued on next page*

*Figure 4 continued*

rate ($V_f$), tidal volume ($V_T$), metabolism demand ($V_{O2}$), inspiratory duration ($T_I$), expiratory duration ($T_E$), breath cycle duration ($T_{TOT}$), inspiratory flow rate ($V_T/T_I$), and expiratory flow rate ($V_T/T_E$). Linear mixed-effects regression model, ****p < 0.0001, ns: p ≥ 0.05.

The online version of this article includes the following figure supplement(s) for figure 4:

**Figure supplement 1.** Characterization of dynamic patterns of breathing in Dbh-Cre; Vglut2 cKO mice under room air and hypercapnia (5% $CO_2$).

---

metabolism demand ($V_{O2}$), inspiratory duration ($T_I$), expiratory duration ($T_E$), breath cycle duration ($T_{TOT}$), inspiratory flow rate ($V_T/T_I$), and expiratory flow rate ($V_T/T_E$) between Dbh-Cre; Vglut2 cKO mice and controls in any of the three 5 min 10% $O_2$ epochs (*Figure 7A, B* and *Table 5*). These data suggest that Vglut2-based glutamatergic signaling in central NA neurons is not required for the hypoxic ventilatory reflex under 10% $O_2$. For the dynamic patterns of breathing, the SD2 of inspiratory flow rate ($V_T/T_I$) (p = 0.005) and the SD2 of minute ventilation ($V_E$) (p = 0.026) were significantly increased during the first epoch (first 5 min) under the 10% $O_2$ challenge in mutant mice, suggesting that Vglut2-based glutamate may play a role in maintaining breathing regularity during the initial, or reflexive, response to hypoxia (*Figure 7—figure supplement 1A–H*).

## Discussion

Neuron co-transmission of two or more neurotransmitters across excitatory, inhibitory, and neuro-modulatory facets is becoming an increasingly appreciated phenomenon in the central nervous system. Functional interrogation of co-transmission is essential not only to understand the distinct and cooperative roles that multiple signaling molecules may play at the synapse but also to determine the most critical targets for potential therapeutics. To our knowledge, the unopposed paradigm in the field until now has been that vesicular glutamate transporter 2 (Vglut2)-based glutamate transmission from central NA neurons is important in respiratory homeostasis. Anatomically, it has long been appreciated that central NA neurons co-express Vglut2, the major glutamate marker among three glutamate transporters (*DePuy et al., 2013*; *Souza et al., 2022b*; *Stornetta et al., 2002a*; *Stornetta et al., 2002b*; *Yang et al., 2021*). Also, it has been well documented that NA Vglut2 positive fibers innervate known central respiratory centers or other autonomic centers, that, when perturbed, could potentially disrupt respiratory homeostasis (*Supplementary file 1*). Functionally, multiple reports have suggested that Vglut2-based glutamate transmission plays a role in respiratory homeostasis (*Abbott et al., 2014*; *Malheiros-Lima et al., 2022*, 2020, 2018). It has also been suggested that, at least for C1 neurons, the apparent lack of a plasmalemmal monoamine transporter may attenuate or eliminate NA or adrenergic release from these fibers (*Guyenet et al., 2013*). However, as discussed below, much of the functional evidence supporting a role for Vglut2-based glutamate transmission from

**Table 1.** Respiratory and metabolic values under room air conditions and the statistical tests.

| | Dbh-Cre; Vglut2 cKO Mean ± SEM | Control Mean ± SEM | Dbh-Cre; Vglut2 cKO vs. Control linear mixed-effects regression model |
|---|---|---|---|
| $V_E/V_{O2}$ | 30.01 ± 2.58 | 33.22 ± 3.97 | p = 0.98 |
| $V_f$ (Breaths/min) | 204.08 ± 10.04 | 213.58 ± 15.36 | p = 0.98 |
| $V_T$ (ml/Breath/g) | 0.009 ± 0.0007 | 0.009 ± 0.0006 | p = 0.88 |
| $V_E$ (ml/min/g) | 1.93 ± 0.14 | 1.98 ± 0.22 | p = 0.995 |
| $V_{O2}$ (ml/min/g) | 0.067 ± 0.004 | 0.061 ± 0.004 | p = 0.78 |
| $T_I$ (s) | 0.117 ± 0.004 | 0.118 ± 0.005 | p = 0.9999 |
| $T_E$ (s) | 0.265 ± 0.020 | 0.255 ± 0.021 | p = 0.996 |
| $T_{TOT}$ (s) | 0.381 ± 0.022 | 0.373 ± 0.024 | p = 0.99997 |
| $V_T/T_I$ (ml/Breath/g/s) | 2.12 ± 0.16 | 2.33 ± 0.26 | p = 0.996 |
| $V_T/T_E$ (ml/Breath/g/s) | 1.43 ± 0.19 | 1.85 ± 0.39 | p = 0.99 |

**Table 2.** Respiratory and metabolic values under 5% $CO_2$ conditions and the statistical tests.

| | Dbh-Cre; Vglut2 cKO Mean ± SEM | Control Mean ± SEM | Dbh-Cre; Vglut2 cKO vs. Control linear mixed-effects regression model |
|---|---|---|---|
| $V_E/V_{O2}$ | 47.99 ± 3.28 | 58.60 ± 5.44 | p = 0.33 |
| $V_f$ (Breaths/min) | 246.72 ± 9.78 | 261.66 ± 14.00 | p = 0.24 |
| $V_T$ (ml/Breath/g) | 0.011 ± 0.0007 | 0.010 ± 0.0008 | p = 0.81 |
| $V_E$ (ml/min/g) | 2.67 ± 0.20 | 2.80 ± 0.32 | p = 0.98 |
| $V_{O2}$ (ml/min/g) | 0.057 ± 0.003 | 0.051 ± 0.005 | p = 0.42 |
| $T_I$ (s) | 0.119 ± 0.004 | 0.118 ± 0.007 | p = 0.62 |
| $T_E$ (s) | 0.135 ± 0.005 | 0.136 ± 0.007 | p = 0.42 |
| $T_{TOT}$ (s) | 0.255 ± 0.009 | 0.254 ± 0.013 | p = 0.32 |
| $V_T/T_I$ (ml/Breath/g/s) | 2.32 ± 0.21 | 2.45 ± 0.25 | p = 0.99 |
| $V_T/T_E$ (ml/Breath/g/s) | 2.17 ± 0.21 | 2.27 ± 0.25 | p = 0.91 |

central NA neurons in breathing is circumstantial or of a non-physiological nature, and absent or attenuated release of adrenaline and noradrenaline has not been demonstrated. Nonetheless, the dominant perspective that NA-Vglut2 glutamate transmission plays a role in breathing has remained unchallenged. Our studies, in contrast to prior work, show that loss of Vglut2 in NA neurons does not appreciably change baseline or chemosensory breathing in unanesthetized and unrestrained mice. Additionally, our work uncovers a novel dynamic expression pattern for Vglut2 and an entirely unde-scribed co-expression domain for Vglut3 in central NA neurons.

Vglut2 has been shown to be co-expressed in subsets of central NA neurons, however, to our knowledge, the potential for the central NA neurons to express the other glutamate transporters Vglut1 or Vglut3 has not been reported at the anatomical level. To determine if central NA neurons express either Vglut1 or Vglut3 and to confirm the Vglut2 expression, we carried out cumulative inter-sectional fate mapping using $Slc17a7^{Cre}$; $Dbh^{p2a-Flpo}$; $Rosa26^{RC::FLTG}$, $Slc17a6^{Cre}$; $Dbh^{p2a-Flpo}$; $Rosa26^{RC::FLTG}$ and $Slc17a8^{Cre}$; $Dbh^{p2a-Flpo}$; $Rosa26^{RC::FLTG}$ compound mouse lines. As this approach maps the cumulative history of expression, we also examined acute adult expression of $Slc17a7$ (Vglut1), $Slc17a6$ (Vglut2), and $Slc17a8$ (Vglut3) using fluorescent RNA in situ hybridization. Our in situ results for $Slc17a6$ (Vglut2) agreed with earlier published outcomes (**DePuy et al., 2013**; **Stornetta et al., 2002a**; **Stornetta et al., 2002b**) showing as much as 80% expression in posterior NA neurons. However, there was no appreciable signal in anterior groups (A7, LC, A5, sub CD/CV). This lack of expression was in stark contrast to the cumulative fate map for Vglut2, showing as much as 75% of the anterior central NA neurons were recombined by the Vglut2-Cre over the lifetime of the animal. To further interrogate the acute activity of the Vglut2-Cre in the Locus Coeruleus in the adult mouse, we injected a Cre respon-sive virus in LC of $Slc17a6^{Cre}$ mice and found sparse Vglut2-Cre positive NA neurons in the LC. The outcomes from the cumulative fate map and acute injections are in agreement with published results from **Yang et al., 2021**. Notably, our use of the $Dbh^{p2a-Flpo}$ avoided expression in the parabrachial nucleus, which was a confounding variable in the $Th^{Flpo}$ mouse. In addition, 28.5% of A5 neurons were shown to be $Slc17a6$ (Vglut2) positive in adult rats by RNA scope (**Souza et al., 2022a**). It is notable that both our lab and the work published by **Stornetta et al., 2002a**; **Stornetta et al., 2002b** failed to detect $Slc17a6$ (Vglut2) expression in anterior groups, but that apparently very low levels can be detected by more sensitive recombinase and RNA scope methods. Additional electrophysiological evidence suggests that this expression is functional (**Yang et al., 2021**). However, it is notable that we see nearly 80% of the LC and 85% of A5 recombined in the intersectional strategy representing life-time expression, but our study and others find only 20% of the LC and 28.5% of A5 may be expressing $Slc17a6$ (Vglut2) at any given moment in adults. Thus, it suggests that Vglut2 co-expression in anterior NA groups is time dependent, though the underlying temporal dynamics of Vglut2 remain unknown. We hypothesize that this could be either a temporal restriction in expression as the animal matures or may reflect dynamic change driven by behavioral or physiological experiences, which could be tested with the use of inducible conditional strategies.

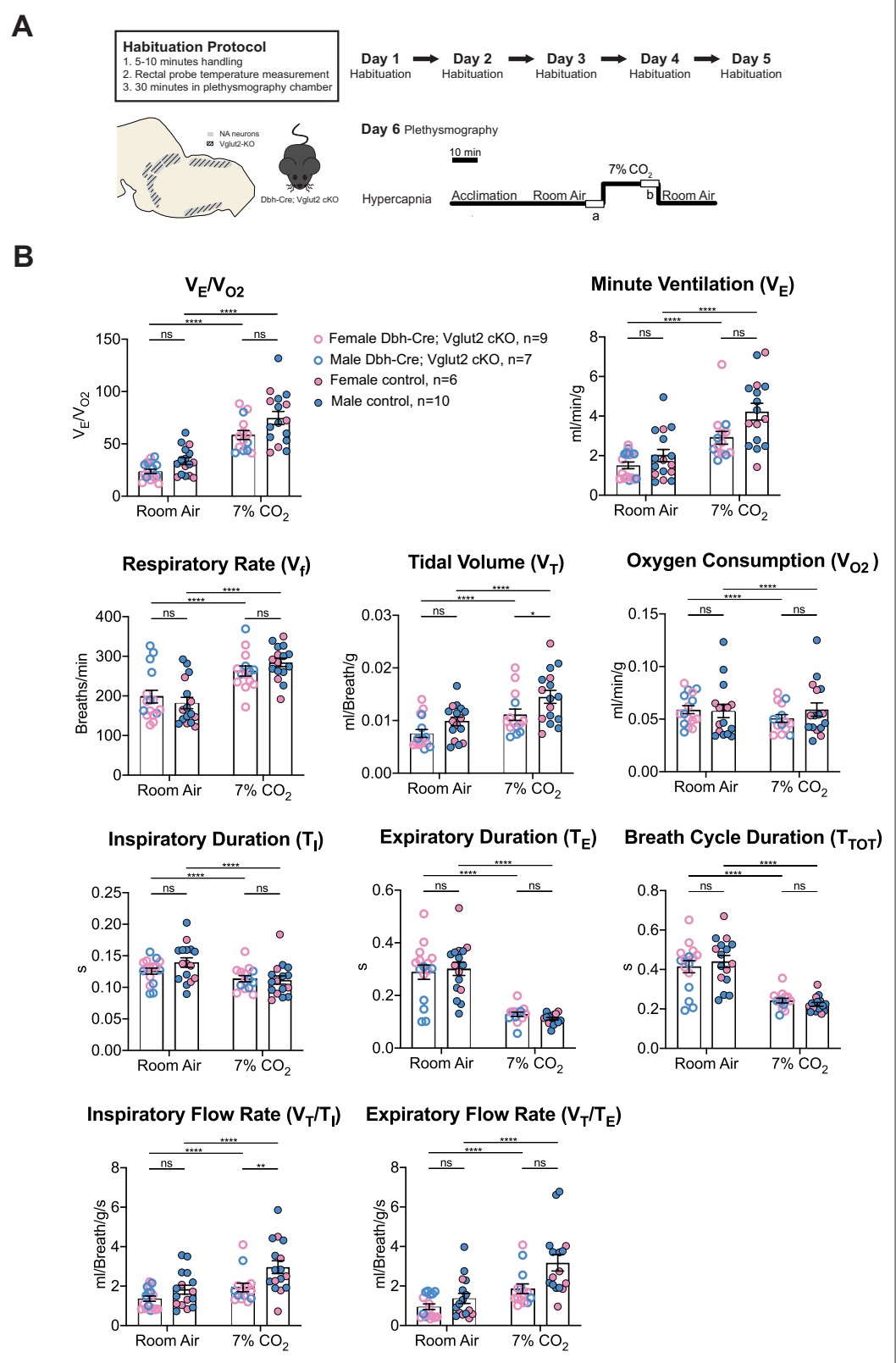

**Figure 5.** Vglut2 conditional knockout in central noradrenergic (NA) neurons fails to alter the majority of hypercapnic ventilation parameters and metabolism under 7% $CO_2$. (**A**) Mouse model schematic and experimental protocol including habituation and hypercapnia protocol (7% $CO_2$). (**B**) Knocking out Vglut2 in the whole NA

*Figure 5 continued on next page*

*Figure 5 continued*

system failed to alter $V_E/V_{O2}$, $V_f$, $V_E$, $V_{O2}$, $T_I$, $T_E$, $T_{TOT}$, $V_T/T_E$ but only showed a significantly reduce in $V_T$ and $V_T/T_I$ under 7% $CO_2$ condition. Linear mixed-effects regression model, ****p < 0.0001, **p < 0.01, *p < 0.05, ns: p ≥ 0.05.

The online version of this article includes the following figure supplement(s) for figure 5:

**Figure supplement 1.** Characterization of dynamic patterns of breathing in Dbh-Cre; Vglut2 cKO mice under room air and hypercapnia (7% $CO_2$).

For *Slc17a7* (Vglut1), neither fate map nor in situ hybridization revealed any co-expression within the central NA system. However, *Slc17a8* (Vglut3) expression was found in the posterior part of NA neurons including C2/A2 and C1/A1. This study is the first to characterize this expression. The majority of this Vglut3 co-expression is restricted to the posterior part of the C2/A2 NA population. This NA region has not been heavily implicated in respiratory control nonetheless it readily lends itself to inter-rogation by intersectional genetic methods and in situ hybridization. Given its distal expression from anterior C1, it is less likely that Vglut3 would compensate for loss of Vglut2 in anterior C1, though this has not been ruled out. Furthermore, while Vglut3 is a marker of glutamatergic neurons, its role in neurotransmission is not wholly defined as Vglut3 is typically found in soma and dendrites (*Fremeau et al., 2004*).

The cumulative fate map and in situ hybridization experiments were performed in both females and males and no obvious sex difference was observed in the expression pattern of Vglut1, Vglut2, or Vglut3. However, only three mice were characterized in each dataset, thus, larger sample sizes may capture more subtle outcomes for potential sex differences.

Based on the Vglut2 co-expression in NA neurons and their projections to multiple brain regions important in respiration, several studies suggest that glutamate in NA populations is important in breathing control. However, the evidence those studies provided is circumstantial, indirect, or limited due to experimental caveats or technical limitations. *Abbott et al., 2014*, using exactly the same mouse model as this study, showed that conditionally knocking out Vglut2 in the whole central NA system did not significantly change respiratory rate under room air, which is consistent with our findings here. Our results differ in that they reported removing Vglut2 from central NA neurons atten-uated an increase in respiratory rate resulting from unilateral C1 optogenetic stimulation. However, while the optogenetic stimulation of C1 neurons did drive a Vglut2-dependent increase in respiratory rate, it is not clear if the resulting change in breathing reflects a native feature of the network that might be engaged in chemosensory responses or other aspects of breathing. It remains possible that strong unilateral optogenetic stimulation (up to 20 Hz) of a small, isolated population drives a Vglut2-dependent, but ectopic, function that is not typically engaged in the normal operation of the breathing network. Such ectopic outcomes could come from several potential, non-exclusive

**Table 3.** Respiratory and metabolic values under 7% $CO_2$ conditions and the statistical tests.

| | Dbh-Cre; Vglut2 cKO Mean ± SEM | Control Mean ± SEM | Dbh-Cre; Vglut2 cKO vs. Control linear mixed-effects regression model |
|---|---|---|---|
| $V_E/V_{O2}$ | 58.48 ± 4.29 | 74.70 ± 6.22 | p = 0.23 |
| $V_f$ (Breaths/min) | 262.69 ± 12.82 | 284.44 ± 10.59 | p = 0.98 |
| $V_T$ (ml/Breath/g) | 0.011 ± 0.0011 | 0.015 ± 0.0012 | p = 0.023* |
| $V_E$ (ml/min/g) | 2.90 ± 0.32 | 4.23 ± 0.43 | p = 0.16 |
| $V_{O2}$ (ml/min/g) | 0.051 ± 0.004 | 0.059 ± 0.006 | p = 0.64 |
| $T_I$ (s) | 0.114 ± 0.005 | 0.112 ± 0.007 | p = 0.90 |
| $T_E$ (s) | 0.128 ± 0.008 | 0.111 ± 0.005 | p = 0.69 |
| $T_{TOT}$ (s) | 0.242 ± 0.012 | 0.223 ± 0.009 | p = 0.81 |
| $V_T/T_I$ (ml/Breath/g/s) | 1.93 ± 0.22 | 2.96 ± 0.33 | p = 0.0016** |
| $V_T/T_E$ (ml/Breath/g/s) | 1.86±0.25 | 3.18 ± 0.41 | p = 0.075 |

**p < 0.01, *p < 0.05.

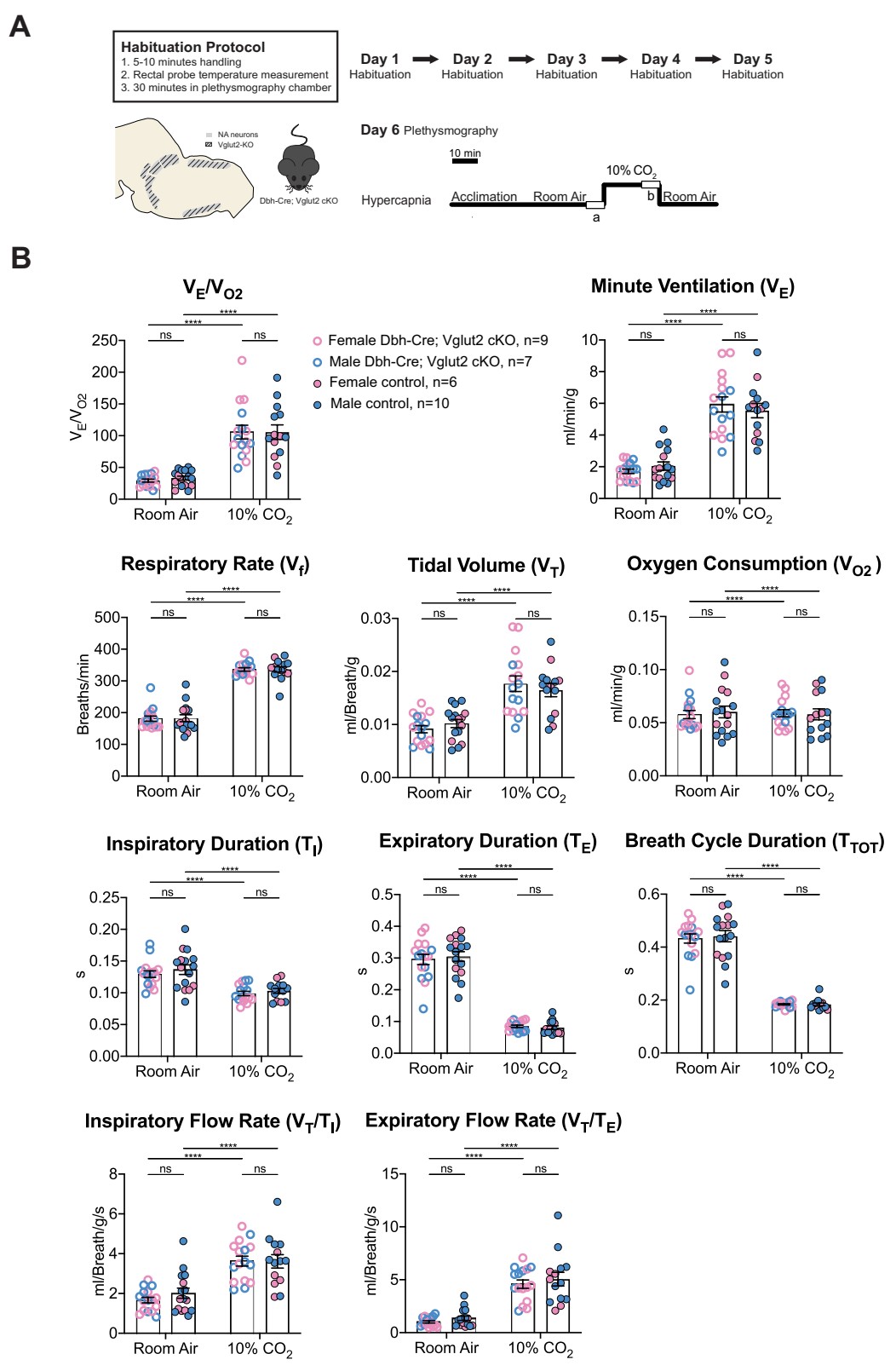

**Figure 6.** Vglut2 conditional knockout in central noradrenergic (NA) neurons fails to alter the hypercapnic ventilatory reflex (10% $CO_2$). (**A**) Mouse model schematic and experimental protocol including habituation and hypercapnia protocol (10% $CO_2$). (**B**) Knocking out Vglut2 in the whole NA system did not significantly alter $V_E/V_{O2}$,

*Figure 6 continued on next page*

*Figure 6 continued*

$V_f$, $V_T$, $V_E$, $V_{O2}$, $T_I$, $T_E$, $T_{TOT}$, $V_T/T_I$, and $V_T/T_E$ under 10% $CO_2$ condition. Linear mixed-effects regression model, ****p < 0.0001, ns: p ≥ 0.05.

The online version of this article includes the following figure supplement(s) for figure 6:

**Figure supplement 1.** Characterization of dynamic patterns of breathing in Dbh-Cre; Vglut2 cKO mice under room air and hypercapnia (10% $CO_2$).

mechanisms. First, unilateral stimulation could drive a network hysteresis or dysregulation similar to a focal injury that is not a typical biological feature (*Ducros et al., 2003*). This point is well made in *Abbott et al., 2013* in their discussion of C1 optogenetic stimulation where they state, 'resulting cardiorespiratory response pattern should not be considered strictly "physiological," because a physiological response is never initiated by selective activation of a single cluster of CNS neurons and the various subsets of C1 neurons are presumably never recruited en bloc under any physiological condition'. Second, strong stimulation could result in overwhelming synaptic mechanisms (i.e., glial uptake) resulting in glutamate spillover and unintended cross talk to affect extra-synaptic neurons proximal to targeted NA fibers. Activity-dependent spillover transmission has been documented elsewhere in the nervous system (*Henneberger et al., 2020*; *Hülsmann et al., 2000*). Third, the outcomes could be secondary to a different autonomic or behavioral function requiring NA-derived glutamatergic signaling involving metabolism or circulation (*DePuy et al., 2013*; *Yang et al., 2021*). Additionally, respiratory parameters except for respiratory rate ($V_f$), such as tidal volume ($V_T$) and minute ventilation ($V_E$) were not reported, making comprehensive comparison difficult.

More circumstantially, *Malheiros-Lima et al., 2020* suggested C1 neurons potentially release glutamate at the pFRG site to regulate active expiration under hypoxia in anesthetized rats by providing three indirect lines of evidence: (1) Vglut2-expressing C1 neurons project to the pFRG region; (2) Increased abdominal expiratory nerve activity ($Abd_{EMG}$) was blunted after blockade of ionotropic glutamatergic receptors at the pFRG site under anesthesia using cytotoxic potassium cyanide (KCN)-mediated hypoxia; and (3) Depletion of C1 neurons eliminated the increased $Abd_{EMG}$ elicited by hypoxia. However, (1) the glutamatergic signaling targeting the pFRG region is not necessarily from C1 NA neurons since glutamatergic neurons from other regions also project to the pFRG site (*Yang and Feldman, 2018*). Thus, it is possible that the glutamatergic signaling that is required for hypoxic response is derived from other glutamatergic neurons such as the preBötzinger complex and the RTN itself. Furthermore, (2) C1 neuron depletion knocks out all the signaling modalities in the C1 population including both noradrenaline and glutamate. It is not clear which signaling drives the hypoxic response. Finally, (3) KCN injection mimics a hypoxic challenge by activating peripheral chemoreceptors, but the physiological responses differ from those seen during environmental hypoxia exposure. Thus, the abdominal activity changes due to KCN injection may not be recaptured under physiological hypoxic challenge.

**Table 4.** Respiratory and metabolic values under 10% $CO_2$ conditions and the statistical tests.

| | Dbh-Cre; Vglut2 cKO Mean ± SEM | Control Mean ± SEM | Dbh-Cre; Vglut2 cKO vs. Control linear mixed-effects regression model |
|---|---|---|---|
| $V_E/V_{O2}$ | 105.77 ± 10.80 | 105.69 ± 11.51 | p = 0.9996 |
| $V_f$ (Breaths/min) | 336.31 ± 5.26 | 336.38 ± 8.52 | p = 0.9997 |
| $V_T$ (ml/Breath/g) | 0.018 ± 0.0015 | 0.016 ± 0.0012 | p = 0.98 |
| $V_E$ (ml/min/g) | 5.94 ± 0.48 | 5.54 ± 0.45 | p = 0.9996 |
| $V_{O2}$ (ml/min/g) | 0.059 ± 0.003 | 0.058 ± 0.005 | p = 0.9999999 |
| $T_I$ (s) | 0.099 ± 0.003 | 0.103 ± 0.004 | p = 0.97 |
| $T_E$ (s) | 0.085 ± 0.004 | 0.081 ± 0.006 | p = 0.63 |
| $T_{TOT}$ (s) | 0.184 ± 0.003 | 0.184 ± 0.005 | p = 0.995 |
| $V_T/T_I$ (ml/Breath/g/s) | 3.62 ± 0.25 | 3.61 ± 0.34 | p = 0.9995 |
| $V_T/T_E$ (ml/Breath/g/s) | 4.59 ± 0.39 | 5.06 ± 0.65 | p = 0.86 |

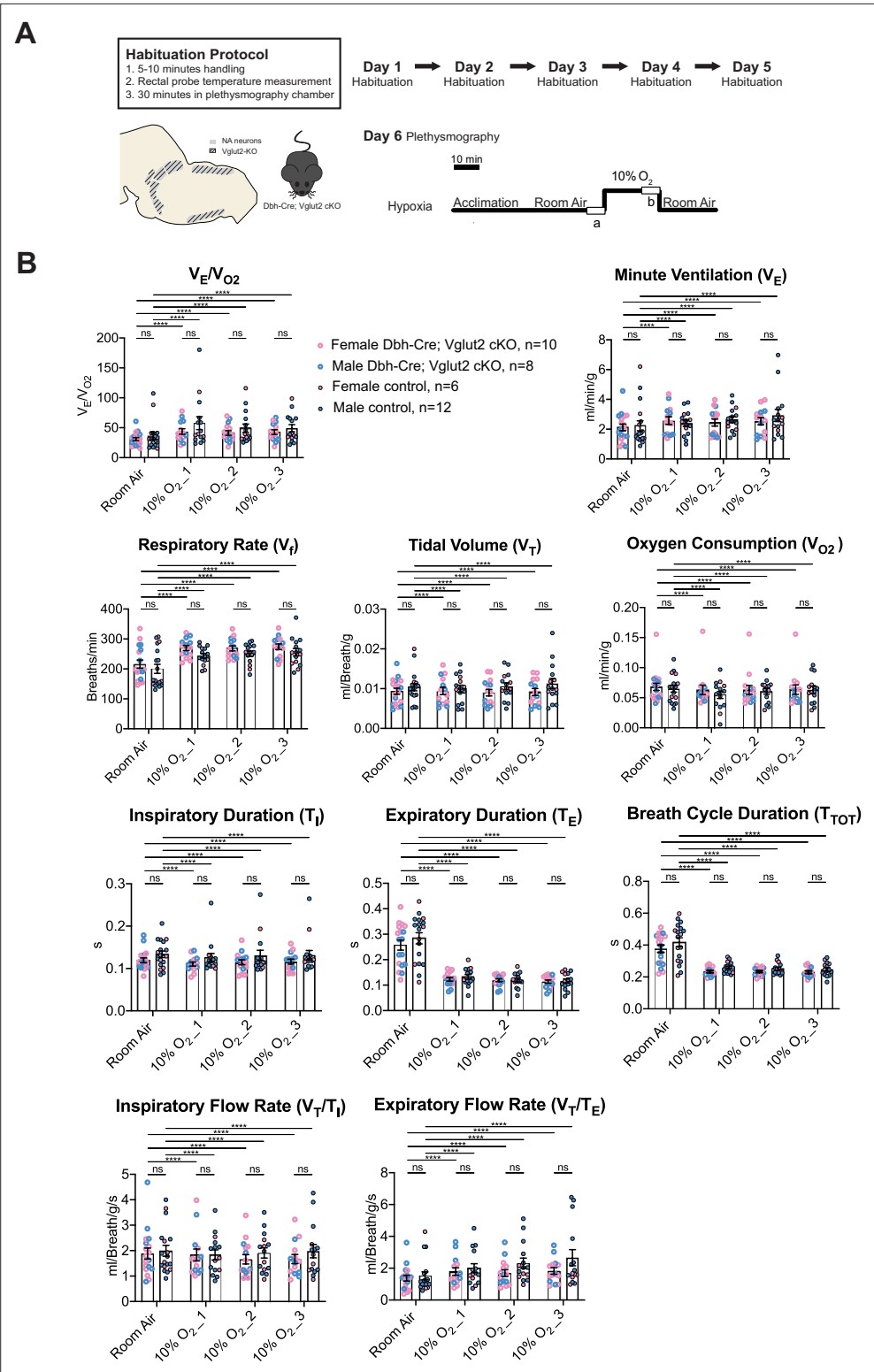

**Figure 7.** Vglut2 conditional knockout in central noradrenergic (NA) neurons fails to alter the hypoxic ventilatory reflex (10% $O_2$). (**A**) Mouse model schematic and experimental protocol including habituation and hypoxia protocol (10% $O_2$). (**B**) Knocking out Vglut2 in the whole NA system did not show significant breathing changes including $V_E/V_{O2}$, $V_E$, $V_f$, $V_T$, $V_{O2}$, $T_I$, $T_E$, $T_{TOT}$, $V_T/T_I$, and $V_T/T_E$ in either of the three 5-min time periods under hypoxia (10% $O_2$). Linear mixed-effects regression model, ****$p < 0.0001$, ns: $p \geq 0.05$.

*Figure 7 continued on next page*

*Figure 7 continued*

The online version of this article includes the following figure supplement(s) for figure 7:

**Figure supplement 1.** Characterization of dynamic patterns of breathing in Dbh-Cre; Vglut2 cKO mice under room air and hypoxia (10% $O_2$).

Similarly, *Malheiros-Lima et al., 2022* and *Malheiros-Lima et al., 2018* showed that Vglut2-expressing C1 neurons project to both the NA A5 region and the preBötzinger complex. The blockade of ionotropic glutamatergic receptors at the A5 region or preBötzinger complex reduced the increase in phrenic nerve activity and the breathing frequency caused by optogenetic stimulation of C1 cells in an anesthetized preparation. Together these results suggest that Vglut2-expressing C1 neurons communicate with A5 or preBötzinger complex neurons by releasing glutamate in turn to regulate phrenic nerve activity or breathing frequency. Again, these two studies showed that C1 neurons are important to regulate phrenic nerve activity or breathing frequency, however, it is not necessarily through a direct C1–A5 or C1–preBötzinger complex glutamatergic pathway, whereas an indirect multi-synaptic pathway originating with C1 noradrenaline and ending with glutamate transmission from an intermediate to A5 or preBötzinger complex cannot be ruled out. Importantly, in all four studies noted here, respiratory measurements were limited or proxies of breathing under anesthesia were used. The role of NA-Vglut2 signaling was not reported in the context of an unanesthetized, unrestrained animal under normoxic, hypoxic, or hypercapnic breathing in these previous studies.

Thus, the limitations and indirect lines of the evidence in these studies raised the question as to whether or not glutamatergic signaling in central NA neurons is required or necessary to modulate respiratory homeostasis under more physiologically relevant circumstances and if NA-derived glutamate is an important mechanism to understand the pathophysiology of NA neurons in respiratory diseases. The goal of our study was to test the requirement of NA-based glutamatergic signaling in breathing homeostasis in the unanesthetized and unrestrained animal under conditions that would engage the whole respiratory control network rather than a singular component that may drive hysteresis or artifactual outcomes. Our in vivo breathing data under room air, hypoxic, and multiple hypercapnic conditions failed to show the requirement of NA-derived Vglut2 in normal breathing and respiratory chemoreflexes directly. For the dynamic patterns of breathing, we measured the SD1 and SD2 of eight parameters under each gas condition. We found no consistent change across multiple hypercapnic challenges or hypoxia. We did find that under severe hypercapnic challenge (10% $CO_2$), NA-derived Vglut2 removal resulted in an increase in the SD1 of breath cycle duration and respiratory

**Table 5.** Respiratory and metabolic values under 10% $O_2$ conditions and the statistical tests.

| | Dbh-Cre; Vglut2 cKO Mean ± SEM | Control Mean ± SEM | Linear mixed-effects regression model | Dbh-Cre; Vglut2 cKO Mean ± SEM | Control Mean ± SEM | Linear mixed-effects regression model | Dbh-Cre; Vglut2 cKO Mean ± SEM | Control Mean ±SEM | Linear mixed-effects regression model |
|---|---|---|---|---|---|---|---|---|---|
| | 10% $O_2$_1 | | | 10% $O_2$_2 | | | 10% $O_2$_3 | | |
| $V_E/V_{O2}$ | 43.79 ± 4.57 | 57.75 ± 10.88 | p=0.99 | 41.30 ± 3.85 | 49.36±6.93 | p=0.99 | 42.82±3.97 | 49.19±6.02 | p=0.99 |
| $V_f$ (Breaths/min) | 269.71 ± 8.49 | 243.89 ± 7.64 | p=0.67 | 268.38 ± 8.40 | 251.15 ± 8.67 | p=0.83 | 273.96 ± 9.31 | 255.84 ± 12.19 | p=0.80 |
| $V_T$ (ml/Breath/g) | 0.009 ± 0.0009 | 0.010 ± 0.0009 | p=0.89 | 0.009 ± 0.0008 | 0.011 ± 0.0009 | p=0.85 | 0.009 ± 0.0009 | 0.011 ± 0.0013 | p=0.64 |
| $V_E$ (ml/min/g) | 2.58 ± 0.25 | 2.40 ± 0.21 | p=0.999998 | 2.45 ± 0.24 | 2.65 ± 0.21 | p=0.9998 | 2.53 ± 0.24 | 2.92 ± 0.41 | p=0.998 |
| $V_{O2}$ (ml/min/g) | 0.063 ± 0.008 | 0.055 ± 0.006 | p=0.995 | 0.063 ± 0.007 | 0.061 ± 0.005 | p=0.9998 | 0.064 ± 0.008 | 0.063 ± 0.006 | p=0.99999999 |
| $T_I$ (s) | 0.111 ± 0.005 | 0.126 ± 0.010 | p=0.44 | 0.115 ± 0.006 | 0.131 ± 0.012 | p=0.51 | 0.116 ± 0.006 | 0.132 ± 0.011 | p=0.44 |
| $T_E$ (s) | 0.123 ± 0.007 | 0.133 ± 0.009 | p=0.995 | 0.119 ± 0.006 | 0.121 ± 0.008 | p=0.9999 | 0.114 ± 0.007 | 0.117 ± 0.008 | p=0.9997 |
| $T_{TOT}$ (s) | 0.234 ± 0.007 | 0.259 ± 0.009 | p=0.53 | 0.234 ± 0.007 | 0.252 ± 0.009 | p=0.75 | 0.230 ± 0.008 | 0.248 ± 0.011 | p=0.69 |
| $V_T/T_I$ (ml/Breath/g/s) | 1.83 ± 0.23 | 1.83 ± 0.20 | p=0.9999 | 1.66 ± 0.19 | 1.91 ± 0.20 | p=0.998 | 1.67 ± 0.18 | 1.98 ± 0.26 | p=0.99 |
| $V_T/T_E$ (ml/Breath/g/s) | 1.79 ± 0.24 | 2.00 ± 0.29 | p=0.9994 | 1.69 ± 0.21 | 2.32 ± 0.32 | p=0.99 | 1.83 ± 0.21 | 2.67 ± 0.51 | p=0.98 |

rate, which was not observed under 5% and 7% $CO_2$ conditions. Under hypoxia (10% $O_2$), Dbh-Cre; Vglut2 cKO mice only showed an increase in the SD2 of inspiratory flow rate and minute ventilation during the first 5 min of 10% $O_2$ exposure. These findings may suggest that Vglut2-based glutamatergic signaling in central NA neurons may play a role in modulating breathing regularity under high $CO_2$ challenge and hypoxia. However, the phenotypes were of a small magnitude and not consistent across challenges, and not at all seen under homeostatic breathing.

To ensure a rigorous and robust conclusion, we included several extra precautions and additional measures in our experimental design and analysis. First, we use exactly the same mouse model, DBH-Cre; Vglut2 cKO, as *Abbott et al., 2014* used, and confirmed, by a second method (fluorescent RNA in situ hybridization) that Vglut2 expression was indeed abrogated in the whole central NA system. Our results are in strong agreement with *DePuy et al., 2013* that recombination is efficient. Notably, we assayed for expression at the cell body, rather than the fibers and see no singular cell body that is scored positive. Second, we measured the breathing of unanesthetized, unrestrained, and habituated mice under physiological challenges including normoxia, hypercapnia, and hypoxia. Third, to visualize breathing function comprehensively, the metabolic parameter and multiple respiratory parameters (both the steady state and the dynamic patterns) were measured and reported in absolute values including oxygen consumption, respiratory rate, tidal volume, minute ventilation, overall respiratory output (minute ventilation normalized to oxygen consumption), inspiratory duration, expiratory duration, breath cycle duration, inspiratory flow rate, and expiratory flow rate. Fourth, our experimental design was overpowered (i.e., *n* = 16–21 vs. 5–13 needed for power based on power analysis and *n* = 7–8 reported in other manuscripts). Each respiratory/metabolic parameter for each physiological condition including room air, hypercapnia, and hypoxia was derived by quantifying the entire respiratory trace and analyzed between mutant and sibling controls by using a linear mixed-effects regression model with animal type (experimental vs. control) as fixed effects, animal ID as a random effect, and sex as an experimental covariate (for which we did not find a significant effect) (*Lusk et al., 2023a*).

Despite our experimental design and the extensive nature of our measurements, there remain several possibilities or considerations in the interpretation of our data. The first possibility is Vglut3 in posterior CA domain (first characterized in this study) compensates for the effect of the loss of Vglut2 in NA neurons in breathing. However, we argue that this is unlikely because (1) Our model is the same used in the other study where a Vglut2-dependent effect was seen with optogenetic stimulation. (2) Vglut3 is expressed in the posterior C2/A2 NA neurons which are anatomically distant from anterior C1 and the LC (the two most likely NA candidates with Vglut2 co-expression to impact breathing). Also, posterior C2/A2 has not been heavily implicated in breathing regulation. Future work will need to further investigate the role of Vglut3 in posterior CA in respiratory control in order to test this possibility. The second possibility is that pre- or post-natal developmental compensation corrects for the loss of NA-derived glutamate. Again, we argue that is unlikely, as (1) we are using the same adult model that provided the most direct evidence and first functionally demonstrated a potential role for C1-derived glutamate in breathing; (2) *Abbott et al., 2014* highlight that C1 NA neurons showed no obvious abnormalities in number, morphology, and projection patterns after knocking out Vglut2; and (3) we have examined the requirement of NA-expressed Vglut2 in P7 neonate mice in the autoresuscitation reflex and saw no differences (data not shown). However, it is still possible that developmental compensation occurs before P7, which is something that could be better tested by using the Dbh-CreERT2 (*Stubbusch et al., 2011*) to more acutely remove Vglut2 expression (though this leaves a window of 2–3 weeks for compensation to occur). Third, as we only address the loss of NA-Vglut2-based signaling across the entire NA system, it remains a formal possibility that this ultimately results in the removal of Vglut2-dependent signaling from two counter balancing regions of the NA system and are therefore left with null results whereas removal from anterior C1 alone would show a requirement. To test this possibility, it would require an intersectional conditional knockout/ loss of function approach, something that has not yet been shown to be efficient and effective and would be beyond the scope of these studies. Fourth, it is possible that another neurotransmitter/ neuropeptide within NA neurons might be released in higher amounts in Dbh-Cre; Vglut2 cKO mice to compensate for the deficiency of glutamate in breathing. Loss of Vglut2 could reduce dopamine release in subsets of dopaminergic neurons (*Alsiö et al., 2011*; *Fortin et al., 2012*; *Hnasko et al., 2010*). Thus, it remains possible that loss of Vglut2 affects dopamine loading into the vesicles and in turn affecting noradrenaline release as dopamine beta hydroxylase (DBH) synthesizes noradrenaline

from dopamine inside the vesicles (*Weinshenker, 2007*). Changes in noradrenaline release in Vglut2 negative NA neurons could be further examined with fast-scan cyclic voltammetry or microdialysis. Neuropeptide Y (NPY) and galanin have been shown to co-exist in LC NA neurons and are involved in regulating energy metabolism, stress, or anxiety (*Ruohonen et al., 2012*; *Tasan et al., 2010*; *Weinshenker and Holmes, 2016*). In our studies, we account for potential metabolic changes that may underlie or be coordinate with changes in breathing. However, we did not test for potential changes in blood pressure or other autonomic functions as well as affective state that may, in some obscure way, compensate for changes in breathing. Further interrogation of NPY or galanin signaling across NA neurons and potential compensatory effects in breathing through changes in metabolism or other autonomic function under stress or anxiety may yield notable insights. Lastly, our results are loss of function in nature. We do not test sufficiency or gain of function and cannot fully rule out a role for NA-Vglut2-based glutamatergic signaling in the control of breathing.

From a translational perspective, our data question whether or not glutamatergic signaling in NA neurons is likely to be a key mechanism and therefore a therapeutic target for breathing disorders, such as Rett Syndrome and SIDS. *Mecp2*-deficient mice (a mouse model of Rett Syndrome that phenocopies many human symptoms) showed both a deficiency of NA populations (reduced number of NA neurons in C2/A2 and C1/A1 group) and highly variable respiratory rhythm at around 4–5 weeks of age (*Roux et al., 2007*; *Viemari et al., 2005*). 80% of Rett Syndrome patients experience breathing issues, such as unstable breathing, episodes of hyperventilation, and breath holds, throughout their lifespan (*Ramirez et al., 2020*). SIDS decedents show NA abnormalities and are hypothesized to ultimately succumb from a failure in cardiorespiratory autoresuscitation (*Chigr et al., 1989*; *Garcia et al., 2013*; *Kopp et al., 1993*; *Mansouri et al., 2001*; *Ozawa et al., 2003*; *Takashima and Becker, 1991*). However, Dbh-Cre; Vglut2 cKO mice have normal baseline and chemosensory respiratory parameters as adults and neonate mice show normal autoresuscitation indicating that perturbed NA-based glutamatergic signaling may not be a key driver in these or other related respiratory pathophysiologies.

In conclusion, our studies show that Vglut2 has a dynamic and extensive expression profile across the central NA system. We were able to characterize a novel Vglut3-expressing NA population in the posterior C2/A2 nuclei. Despite prior studies providing indirect and circumstantial evidence that NA-based glutamatergic signaling may play a role in control of breathing, our conditional loss of function studies in adult unanesthetized and unrestrained mice failed to consistently and significantly change room air breathing, the hypercapnic ventilatory reflex, and the hypoxic ventilatory reflex. These outcomes offer a contrasting perspective from the current field view and provides further insight into the potential role of NA-glutamate transmission in the control of breathing and NA neuron dysfunction in respiratory disorders.

## Materials and methods
### Ethical approval
Studies were approved by the Baylor College of Medicine Institutional Animal Care and Use Committee (IACUC) under protocol AN-6171, and all experiments reported here were performed in accordance with relevant guidelines and regulations.

### Breeding, genetic background, and maintenance of mice
We maintained all our heterozygous mouse strains by backcrossing to wildtype C57BL/6J mice and homozygous mouse strains by sibling crosses. For immunofluorescence experiments, heterozygous B6;129S-*Slc17a7*$^{tm1.1(cre)Hze}$/J (*Slc17a7*$^{Cre}$, Vglut1-Cre) (Jax Stock No: 023527), *Slc17a6*$^{tm2(cre)Lowl}$/J (*Slc17a6*$^{Cre}$, Vglut2-Cre) (Jax Stock No: 016963), and B6;129S-*Slc17a8*$^{tm1.1(cre)Hze}$/J (*Slc17a8*$^{Cre}$, Vglut3-Cre) (Jax Stock No: 018147) were mated with heterozygous *Dbh*$^{em2.1(flpo)Rray}$ (*Dbh*$^{p2a-Flpo}$, DBH-p2a-Flpo) (MMRRC ID: 41575) (*Sun and Ray, 2016*) mice, respectively, to derive compound lines with both Cre and Flpo alleles (*Slc17a7*$^{Cre}$; *Dbh*$^{p2a-Flpo}$, *Slc17a6*$^{Cre}$; *Dbh*$^{p2a-Flpo}$, and *Slc17a8*$^{Cre}$; *Dbh*$^{p2a-Flpo}$). Then these three compound lines were each mated with homozygous B6.Cg-*Gt(ROSA)26Sor*$^{tm1.3(CAG-tdTomato,-EGFP)Pjen}$/J (*Rosa26*$^{RC::FLTG}$) (Jax Stock No: 026932) to derive three different intersectional mouse lines *Slc17a7*$^{Cre}$; *Dbh*$^{p2a-Flpo}$; *Rosa26*$^{RC::FLTG}$, *Slc17a6*$^{Cre}$; *Dbh*$^{p2a-Flpo}$; *Rosa26*$^{RC::FLTG}$, and *Slc17a8*$^{Cre}$; *Dbh*$^{p2a-Flpo}$; *Rosa26*$^{RC::FLTG}$. For in situ hybridization experiments, wildtype C57BL/6J mice were ordered from the Center of Comparative Medicine (CCM), Baylor College of Medicine. For NA-Vglut2 conditional

loss of function in situ hybridization and plethysmography experiments, hemizygous transgene Tg(D-bh-cre)KH212Gsat (Dbh-Cre) (MMRRC ID: 036778-UCD GENSAT) mice were mated with homozygous $Slc17a6^{tm1Lowl}$/J ($Slc17a6^{flox/flox}$) (Jax Stock No: 012898) to derive Dbh-Cre; $Slc17a6^{flox/+}$. Dbh-Cre; $Slc17a6^{flox/+}$ mice were mated with $Slc17a6^{flox/flox}$ to derive Dbh-Cre; $Slc17a6^{flox/flox}$ (Dbh-Cre; Vglut2 cKO). Sibling mice that lacked the Cre allele or carried the Cre allele but lacked the floxed Vglut2 alleles were used as controls. Rosa26-specific primers for the $Rosa26^{RC::FLTG}$ mice were 5′-GCACTTGC TCTCCCAAAGTC, 5′-GGGCGTACTTGGCATATGAT, and 5′-CTTTAAGCCTGCCCAGAAGA (**Ray et al., 2011**) and yield a 495-bp band (targeted) and 330-bp band (wt). Cre-specific primers for all Cre drivers were 5′-ATCGCCATCTTCCAGCAGGCGCACCATTGCCC and 5′-GCATTTCTGGGGATTGCTTA and yielded a 550-bp band if positive. Flpo-specific primers for $Dbh^{p2a-Flpo}$ are 5′-CACGCCCAGGTA CTTGTTCT and 5′-CCACAGCAAGAAGATGCTGA (**Sun and Ray, 2016**) and yielded a 226-bp band if positive. $Slc17a6^{flox}$-specific primers for Vglut2-floxed mice are available at The Jackson Laboratory website (https://www.jax.org/strain/012898).

## Immunofluorescence staining

$Slc17a7^{Cre}$; $Dbh^{p2a-Flpo}$; $Rosa26^{RC::FLTG}$, $Slc17a6^{Cre}$; $Dbh^{p2a-Flpo}$; $Rosa26^{RC::FLTG}$, and $Slc17a8^{Cre}$; $Dbh^{p2a-Flpo}$; $Rosa26^{RC::FLTG}$ adult mice of both sexes were sacrificed and transcardially perfused with 0.1 M phosphate-buffered saline (PBS) then with 4% paraformaldehyde (PFA) in PBS. Mouse brains were dissected out and fixed for 2 hr in 4% PFA before a PBS rinse and dehydration in 30% sucrose in PBS. Brains were embedded in OCT blocks, sectioned at the thickness of 30 µm, mounted on slides, and stored at −80°C until they were ready for staining. The slides were hydrated in 0.1% Triton-X in PBS (PBST) for 15 min, blocked with 5% donkey serum in 0.1% PBST for 1 hr at room temperature and then incubated with primary antibodies for 72 hr at 4°C in 0.1% PBST with 5% donkey serum. Tissues were washed in 0.1% PBST three times for 10 min each and then incubated with secondary antibodies for 2 hr at room temperature in 0.1% PBST with 5% donkey serum. Slides were washed with 0.1% PBST for 10 min and washed in PBS twice for 10 min each, stained for DAPI, washed three times for 10 min each with PBS and mounted in ProLong Glass (Invitrogen). The following primary and secondary antibodies were used: chicken anti-GFP (1:1000, Abcam ab13970), rabbit anti-dsRed (1:1000, Clontech 632496), donkey anti-chicken Cy2 (1:500, Jackson 703-225-155), and donkey anti-rabbit Cy3 (1:500, Jackson 711-165-152).

## In situ hybridization

Mouse brains were dissected out from adult mice of both sexes with 6–8 weeks of age, sectioned into 25 µm brain sections and mounted on slides. We generated a digoxigenin (DIG)-labeled mRNA antisense probe against $Slc17a7$ (Vglut1), $Slc17a6$ (Vglut2), and $Slc17a8$ (Vglut3), and fluorescein (FITC)-labeled mRNA against $Dbh$ using reverse-transcribed mouse cDNA as a template and an RNA DIG or FITC-labeling kits from Roche (Sigma). Primer and probe sequences for the $Slc17a7$ (Vglut1), $Slc17a6$ (Vglut2), and $Slc17a8$ (Vglut3) and $Dbh$ probes are available in the Allen Brain Atlas (http://www.brain-map.org). For the $Slc17a6$ (Vglut2) and $Dbh$ double ISH in Dbh-Cre; Vglut2 cKO and their littermate controls, we generated a new $Slc17a6$ (Vglut2) probe targeting exon 2 of $Slc17a6$ specifically (**Tong et al., 2007**) and the probe sequence is 892–1144 bp as Slc17a6 transcript variant 1 and the size is 253 bp. ISH was performed by the RNA In Situ Hybridization Core at Baylor College of Medicine using an automated robotic platform as previously described (**Yaylaoglu et al., 2005**) with modifications of the protocol for double ISH. Modifications in brief (see **Yaylaoglu et al., 2005** for buffer descriptions): both probes were hybridized to the tissue simultaneously. After the described washes and blocking steps the DIG-labeled probes were visualized using tyramide-Cy3 Plus (1/75 dilution, 15-min incubation, Akoya Biosciences). After washes in TNT buffer, the remaining horseradish peroxidase (HRP) activity was quenched by a 10-min incubation in 0.2 M HCl. The sections were then washed in TNT, blocked in TNB for 15 min, and incubated at room temperature for 30 min with HRP-labeled sheep anti-FITC antibody (1/500 in TNB, Roche/Sigma). Following washes in TNT, the FITC-labeled probe was visualized using tyramide-FITC Plus (1/50 dilution, 15-min incubation, Akoya Biosciences). Following washes in TNT, the slides were stained with DAPI (invitrogen), washed again, removed from the machine, and mounted in ProLong Diamond (Invitrogen).

## Viral injection

To verify Vglut2 co-expression in LC in adult mice (*Yang et al., 2021*), 6-week-old *Slc17a7$^{Cre}$* mice were injected with pAAV-EF1a-DIO-tdTomato-WPRE virus (RRID:Addgene_133786, obtained from Joshua Ortiz at the Optogenetics and Viral Vectors Core at the Jan and Dan Duncan Neurological Research Institute, 1 ul at 7.90E+10 Gc/ml) into the LC (coordinates from bregma anteroposterior −5.4 mm, lateral +0.8 mm, and dorsoventral −4.0 mm) and allowed to incubate for 4 weeks.

## Plethysmography

Plethysmography on unanesthetized and free-moving mice was carried out as described in *Ray et al., 2011*. Six- to eight-week-old adult mice of both sexes with minimum group sizes *n* = 16 were used in both the experimental and control group for each experiment. Animals were subjected to a 5-day habituation protocol with each day including several minutes of handling, temperature taken by rectal probe and at least 30 min exposure in the plethysmography chamber (*Martinez et al., 2019*). Plethysmography started to be performed on the sixth day and was finished for all animals within a week of the last day of habituation (maximum 6 mice can be assayed for plethysmography per day due to the limited number of plethysmography rigs). On the day of testing, mice were taken out from their home cage, weighed, and rectal temperature was taken. Animals were then placed into a flow-through, temperature-controlled (about 32°C with real-time temperature recording) plethysmography chamber and allowed to acclimate for 20–40 min in room air (21% $O_2$/79% $N_2$) conditions. After acclimation (indicated by a steady respiratory trace free from movement artifact), a 20-min baseline breathing trace was taken under room air, then the chamber gas was switched to a hypercapnic or hypoxic mixture of 5% $CO_2$/21% $O_2$/74% $N_2$, 7% $CO_2$/21% $O_2$/72% $N_2$, 10% $CO_2$/21% $O_2$/69% $N_2$, or 10% $O_2$/90% $N_2$, depending on the protocol, for 20 min. Chamber gas was then switched back to room air for another 20 min. The animals were removed from the chamber and rectal temperature was measured immediately after the mice were taken out from the chamber. Each testing period for an individual mouse was separated by 24 hr to allow for a full recovery.

## Plethysmography data analysis and statistics

Details were previously described in *Martinez et al., 2019*; *Ray et al., 2011*. Plethysmography pressure changes were measured using a Validyne DP45 differential pressure transducer, CD15 carrier demodulator and a reference chamber, and were recorded with LabChart Pro in real time. Respiratory waveforms were analyzed by the SASSI module of the Breathe Easy software to determine respiratory frequency ($V_f$), tidal volume ($V_T$), minute ventilation ($V_E$), oxygen consumption ($V_{O2}$), ventilatory equivalents for oxygen ($V_E/V_{O2}$), inspiratory duration ($T_I$), expiratory duration ($T_E$), breath cycle duration ($T_{TOT}$), inspiratory flow rate ($V_T/T_I$), and expiratory flow rate ($V_T/T_E$) (*Lusk et al., 2023a*). Poincaré plots for $T_I$, $T_E$, $T_{TOT}$, $V_T/T_I$, $V_T/T_E$, $V_f$, $V_T$, $V_E$ and their SD1 and SD2 properties were generated and characterized by the STAGG module of the Breathe Easy software. Analysis windows of the respiratory parameters were as follows: for room air, the entire experimental period was considered; for hypercapnia, only the last 5-min challenge for each of 5/7/10% $CO_2$ conditions were analyzed; and for hypoxia, three separate 5-min time periods, namely the 5th–10th, 10th–15th min, and 15th–20th min intervals, were used due to the stereotypical biphasic respiratory response under this condition. Only steady quiescent breathing periods were included in the data analysis. A power analysis was performed using the reported effect size in *Abbott et al., 2014*, which used the same mouse model as we used here; 5–13 mice were necessary to observe a statistically significant result (*Abbott et al., 2014*) was able to see a significant difference between two groups with *n* = 7 mice. In our experiments, the sample size for each group (Dbh-Cre; Vglut2 cKO and control) exceeded 13. Steady-state results ($V_f$, $V_E$, $V_T$, $V_{O2}$, $V_E/V_{O2}$, $T_I$, $T_E$, $T_{TOT}$, $V_T/T_I$, $V_T/T_E$) for room air and hypercapnic or hypoxic data were compared between Dbh-Cre; Vglut2 cKO cohorts and sibling controls using a linear mixed-effects regression model with animal type (experimental vs. control) as fixed effects and animal ID as a random effect (*Lusk et al., 2023a*). SD1 and SD2 results for room air and hypercapnic or hypoxic data were compared between Dbh-Cre; Vglut2 cKO cohorts and sibling controls using a Mann–Whitney *U* test (*Ferreira et al., 2022*). A p-value threshold of p < 0.05 was used to test for statistical significance. Individual data points, means, and standard errors of the mean are shown on all charts. The graphs were plotted by Prism 8.

## Image quantification

Images were taken by using a Zeiss upright epifluorescent microscope and a Zeiss LSM 880 with Airyscan FAST confocal microscope. Images were captured using Zen software with z-stack function from top to bottom with 0.34 μm intervals, exported, and then analyzed in Imaris using the spots and surface functions. For quantification of immunofluorescence staining, each GFP positive area coincident with DAPI (to denote nuclear localization) was defined as Vglut2 or Vglut3 expressing NA neurons while each tdTomato positive area coincident with DAPI was defined as NA neurons without any Vglut2 or Vglut3 co-expression. For quantification of *Slc17a6* (Vglut2) or *Slc17a8* (Vglut3) with *Dbh* double in situ hybridization in adult WT mice, *Dbh* positive areas coinciding with DAPI were identified as NA neurons and the *Dbh* positive areas overlapped with *Slc17a6* (Vglut2) or *Slc17a8* (Vglut3) positive pixels and DAPI were defined as NA neurons colocalized with *Slc17a6* (Vglut2) or *Slc17a8* (Vglut3). The number of *Slc17a6* (Vglut2) or *Slc17a8* (Vglut3) positive NA neurons and *Slc17a6* (Vglut2) or *Slc17a8* (Vglut3) negative NA neurons was counted in each image for both immunofluorescence and in situ experiments and the percentage of *Slc17a6* (Vglut2) or *Slc17a8* (Vglut3) positive NA neurons among all NA neurons in each NA nucleus in each mouse brainstem was calculated every other brain section unilaterally. For quantification of *Slc17a6* (Vglut2) and *Dbh* double ISH in Dbh-Cre; Vglut2 cKO and their littermate controls, *Slc17a6* (Vglut2) pixel intensities in *Dbh* positive areas coincident with DAPI (NA neurons) were measured in mutant and control images separately. The *Slc17a6* (Vglut2) pixel intensity in NA neurons of control brains was normalized as 1 and the relative *Slc17a6* (Vglut2) pixel intensity of NA neurons in mutant brains compared to that in control brains was calculated. At least three mouse brains were examined for each set of experiments in each group. All quantitative results were graphed using Prism 8.

## Acknowledgements

We thank the Optical Imaging & Vital Microscopy Core (OiVM) at Baylor College of Medicine with the expert assistance of Jason Kirk for confocal imaging. We thank BCM Neuropathology Core and Tao Lin for tissue sectioning. We thank the RNA In Situ Hybridization Core at Baylor College of Medicine with the expert assistance of Cecilia Ljungberg for performing in situ hybridization (NIH S10 OD016167 and NIH IDDRC Grant P50 HD103555). We thank Dr. Joshua Ortiz and Dr. Benjamin Arenkiel at the NRI Optogenetics and Viral Vectors Core for providing Cre-responsive AAV virus used in viral injection.

## Additional information

### Funding

| Funder | Grant reference number | Author |
| --- | --- | --- |
| National Heart, Lung, and Blood Institute | R01HL130249 | Russell S Ray |

The funders had no role in study design, data collection, and interpretation, or the decision to submit the work for publication.

### Author contributions

Yuan Chang, Writing – original draft, Writing – review and editing, Designed and completed immunofluorescence staining, imaging, and plethysmography experiments, Analyzed the data; Savannah Lusk, Writing – review and editing, Completed the viral injection experiments; Andersen Chang, Software, Writing – review and editing, Helped analyze the data; Christopher S Ward, Software, Writing – review and editing, Helped analyze the data; Russell S Ray, Resources, Funding acquisition, Writing – review and editing, Conceptualized and designed the study, Analyzed the data, Helped write the initial manuscript

### Author ORCIDs

Yuan Chang (ID) https://orcid.org/0000-0001-9837-2897
Russell S Ray (ID) https://orcid.org/0000-0001-9610-2703

### Ethics

Studies were approved by the Baylor College of Medicine Institutional Animal Care and Use Committee (IACUC) under protocol AN-6171, and all experiments reported here were performed in accordance with relevant guidelines and regulations.

Reviewer #1 (Public review): https://doi.org/10.7554/eLife.88673.4.sa1
Reviewer #2 (Public review): https://doi.org/10.7554/eLife.88673.4.sa2
Reviewer #4 (Public review): https://doi.org/10.7554/eLife.88673.4.sa3
Author response https://doi.org/10.7554/eLife.88673.4.sa4

## Additional files

### Supplementary files

• Supplementary file 1. Vglut2 positive innervations from central noradrenergic neurons to the brain nuclei important in breathing control.

• MDAR checklist

### Data availability

Source data files have been provided for Figures 1, 2, 3, and their supplements. Raw plethysmography data for Figures 4, 5, 6, 7, and their supplements is posted on Zenodo (https://doi.org/10.5281/zenodo.11557289; https://doi.org/10.5281/zenodo.11557468; https://doi.org/10.5281/zenodo.11557554; https://doi.org/10.5281/zenodo.11557569). Breathe Easy software code used for plethysmography data analysis is posted on Github: https://github.com/MolecularNeurobiology/Breathe_Easy (*Lusk et al., 2023b*).

The following datasets were generated:

| Author(s) | Year | Dataset title | Dataset URL | Database and Identifier |
|---|---|---|---|---|
| Chang Y, Ray R | 2024 | Raw plethysmography data for Figure 4_5% CO2 protocol | https://doi.org/10.5281/zenodo.11557289 | Zenodo, 10.5281/zenodo.11557289 |
| Chang Y, Ray R | 2024 | Raw plethysmography data for Figure 5_7% CO2 protocol | https://doi.org/10.5281/zenodo.11557468 | Zenodo, 10.5281/zenodo.11557468 |
| Chang Y, Ray R | 2024 | Raw plethysmography data for Figure 6_10% CO2 protocol | https://doi.org/10.5281/zenodo.11557554 | Zenodo, 10.5281/zenodo.11557554 |
| Chang Y, Ray R | 2024 | Raw plethysmography data for Figure 7_10% O2 protocol | https://doi.org/10.5281/zenodo.11557569 | Zenodo, 10.5281/zenodo.11557569 |

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
