## [Editor Report · eLife assessment]

Chang et al. provide glutamate co-expression profiles in the central noradrenergic system and test the requirement of Vglut2-based glutamatergic release in respiratory and metabolic activity under physiologically relevant gas challenges. Their experiments provide **compelling** evidence that conditional deletion of vesicular glutamate transporters from noradrenergic neurons does not impact steady-state breathing or metabolic activity in room air, hypercapnia, or hypoxia. This study provides an **important** contribution to our understanding of how noradrenergic neurons regulate respiratory homeostasis in conscious adult mice.

---

## [Referee Report · Reviewer #1 (Public review)]

Summary:

Chang et al. provide glutamate co-expression profiles in the central noradrenergic system and test the requirement of Vglut2-based glutamatergic release in respiratory and metabolic activity under physiologically relevant gas challenges. Their experiments show that conditional deletion of Vglut2 in NA neurons does not impact steady-state breathing or metabolic activity in room air, hypercapnia, or hypoxia. Their observations challenge the importance of glutamatergic signaling from Vglut2 expressing NA neurons in normal respiratory homeostasis in conscious adult mice.

Strengths:

The comprehensive Vglut1, Vglut2, and Vglut3 co-expression profiles in the central noradrenergic system and the combined measurements of breathing and oxygen consumption are two major strengths of this study. Observations from these experiments provide previously undescribed insights into (1) expression patterns for subtypes of the vesicular glutamate transporter protein in the noradrenergic system and (2) the dispensable nature of Vglut2-dependent glutamate signaling from noradrenergic neurons to breathing responses to physiologically relevant gas challenges in adult conscious mice.

Weaknesses:

Although the cellular expression profiles for the vesicular glutamate transporters are provided, the study does not document that glutamatergic-based signaling originating from noradrenergic neurons is evident at the cellular level under normal, hypoxic, and/or hypercapnic conditions. The authors effectively recognize this issue and appropriately discuss their findings in this context.

---

## [Referee Report · Reviewer #2 (Public review)]

The authors characterized the recombinase-based cumulative fate maps for vesicular glutamate transporters (Vglut1, Vglut2 and Vglut3) expression and compared those maps to their real-time expression profiles in central NA neurons by RNA in situ hybridization in adult mice. Authors have revealed a new and intriguing expression pattern for Vglut2, along with an entirely uncharted co-expression domain for Vglut3 within central noradrenergic neurons. Interestingly, and in contrast to previous studies, the authors demonstrated that glutamatergic signaling in central noradrenergic neurons does not exert any influence on breathing and metabolic control either under normoxic/normocapnic conditions or after chemoreflex stimulation. Also, they showed for the first-time the Vglut3-expressing NA population in C2/A2 nuclei. In addition, they were also able to demonstrate Vglut2 expression in anterior NA populations, such as LC neurons, by using more refined techniques, unlike previous studies.

A major strength of the study is the use of a set of techniques to investigate the participation of NA-based glutamatergic signaling in breathing and metabolic control. The authors provided a full characterization of the recombinase-based cumulative fate maps for Vglut transporters. They performed real-time mRNA expression of Vglut transporters in central NA neurons of adult mice. Further, they evaluated the effect of knocking down Vglut2 expression in NA neurons using a DBH-Cre; Vglut2cKO mice on breathing and control in unanesthetized mice. Finally, they injected the AAV virus containing Cre-dependent Td tomato into LC of v-Glut2 Cre mice to verify the VGlut2 expression in LC-NA neurons. A very positive aspect of the article is that the authors combined ventilation with metabolic measurements. This integration holds particular significance, especially when delving into the exploration of respiratory chemosensitivity. Furthermore, the sample size of the experiments is excellent.

Despite the clear strengths of the paper, some weaknesses exist. It is not clear in the manuscript if the experiments were performed in males and females and if the data were combined. I believe that the study would have benefited from a more comprehensive analysis exploring the sex specific differences. The reason I think this is particularly relevant is the developmental disorders mentioned by the authors, such as SIDS and Rett syndrome, which could potentially arise from disruptions in central noradrenergic (NA) function, exhibit varying degrees of sex predominance. Moreover, some of the noradrenergic cell groups are sexually dimorphic. For instance, female Wistar rats exhibit a larger LC size and more LC-NA neurons than male subjects (Pinos et al., 2001; Garcia-Falgueras et al., 2005). More recently, a detailed transcriptional profiling investigation has unveiled the identities of over 3,000 genes in the LC. This revelation has highlighted significant sexual dimorphisms, with more than 100 genes exhibiting differential expression within LC-NA neurons at the transcript level. Furthermore, this investigation has convincingly showcased that these distinct gene expression patterns have the capacity to elicit disparate behavioral responses between sexes (Mulvey et al., 2018). Therefore, the authors should compare the fate maps, Vglut transporters in males and females, at least considering LC-NA neurons. Even in the absence of identified sex differences, this information retains significant importance.

An important point well raised by the authors is that although suggestive, these experiments do not definitively rule out that NA-Vglut2 based glutamatergic signaling has a role in breathing control. Subsequent experiments will be necessary to validate this hypothesis.

An improvement could be made in terms of measuring body temperature. Opting for implanted sensors over rectal probes would circumvent the need to open the chamber, thereby preventing alterations in gas composition during respiratory measurements. Further, what happens to body temperature phenotype in these animals under different gas exposures? These data should be included in the Tables.

Is it plausible that another neurotransmitter within NA neurons might be released in higher amounts in DBH-Cre; Vglut2 cKO mice to compensate for the deficiency in glutamate and prevent changes in ventilation?

Continuing along the same line of inquiry is there a possibility that Vglut2 cKO from NA neurons not only eliminates glutamate release but also reduces NA release? A similar mechanism was previously found in VGLUT2 cKO from DA neurons in previous studies (Alsio et al., 2011; Fortin et al., 2012; Hnasko et al., 2010). Additionally, does glutamate play a role in the vesicular loading of NA? Therefore, could the lack of effect on breathing be explained by the lack of noradrenaline and not glutamate?

---

## [Referee Report · Reviewer #4 (Public review)]

Summary:

Although previous research suggested that noradrenergic glutamatergic signaling could influence respiratory control, the work performed by Chang and colleagues reveals that excitatory (specifically Vglut2) neurons is dynamically and widely expressed throughout the central noradrenergic system, but it is not significantly crucial to change baseline breathing as well the hypercapnia and hypoxia ventilatory responses. The central point that will make a significant change in the field is how NA-glutamate transmission may influence breathing control and the dysfunction of NA neurons in respiratory disorders.

Strengths:

There are several strengths such as the comprehensive analysis of Vglut1, Vglut2, and Vglut3 expression in the central noradrenergic system and the combined measurements of breathing parameters in conscious unrestrained mice.

Other considerations :

These results strongly suggest that glutamate may not be necessary for modulating breathing under normal conditions or even when faced with high levels of carbon dioxide (hypercapnia) or low oxygen levels (hypoxia). This finding is unexpected, considering many studies have underscored glutamate's vital role in respiratory regulation, more so than catecholamines. This leads us to question the significance of catecholamines in controlling respiration. Moreover, if glutamate is not essential for this function, we need to explore its role in other physiological processes such as sympathetic nerve activity (SNA), thermoregulation, and sensory physiology.

---

## [Author Response]

The following is the authors’ response to the previous reviews.

All of the reviewers indicate that their major concerns have been adequately addressed, but they each have a few comments that the authors should consider before submitting a final version (without further review) for publication. For example, a statement about the sex of the mice used in the studies and whether any differences were noted if both sexes were used. The idea that the loss of glutamate transport might affect NA loading into vesicles is also worth considering. Finally, the authors might want to mention that the role of neuropeptide release from NA neurons needs further examination.

As noted in the prior submitted revision, all experiments contained both males and females and this was addressed in our re-submission. In our analysis of breathing and metabolism, sex was included in the analysis and no significant phenotypic difference was observed (The statement of no sex difference is in line 451-456). For the fate map and in situ experiments, although the group size is small, we did not see obvious differences in the expression patterns in the three glutamate transporters between females and males (line 347-350). All the anatomical and phenotypic data in this manuscript are presented as combined graphs (figure 1, figure 1 supplement 1, figure 2, figure 2 supplement 2, figure 4,5,6,7) and we had differentially labeled our data points by sex (female data is pink and male data is blue).

The possibility that loss of Vglut2 might affect NA release has been added in the discussion (line 485-491) of the current revision. Dopamine Beta Hydroxylase (DBH) converts dopamine to noradrenaline in the vesicles, thus, glutamate may not directly affect noradrenaline loading into vesicles. However, since loss of Vglut2 reduced dopamine release in subsets of dopaminergic neurons, it remains possible that glutamate affects dopamine loading in NA neurons and in turn perturbs DA to NA conversion in the vesicle by DBH and subsequent noradrenaline release. Future work could examine this hypothesis using fast-scan cyclic voltammetry (FSCV) or microdialysis.

The further examination of the role of neuropeptide release from NA neurons is mentioned in the discussion (line 491-494 and line 497-499 of the pre).

**eLife assessment**
Chang et al. provide glutamate co-expression profiles in the central noradrenergic system and test the requirement of Vglut2-based glutamatergic release in respiratory and metabolic activity under physiologically relevant gas challenges. Their experiments provide compelling evidence that conditional deletion of vesicular glutamate transporters from noradrenergic neurons does not impact steady-state breathing or metabolic activity in room air, hypercapnia, or hypoxia. This study provides an important contribution to our understanding of how noradrenergic neurons regulate respiratory homeostasis in conscious adult mice.
**Public Reviews:**

**Reviewer #1 (Public Review):**
Summary:Chang et al. provide glutamate co-expression profiles in the central noradrenergic system and test the requirement of Vglut2-based glutamatergic release in respiratory and metabolic activity under physiologically relevant gas challenges. Their experiments show that conditional deletion of Vglut2 in NA neurons does not impact steady-state breathing or metabolic activity in room air, hypercapnia, or hypoxia. Their observations challenge the importance of glutamatergic signaling from Vglut2 expressing NA neurons in normal respiratory homeostasis in conscious adult mice.Strengths:The comprehensive Vglut1, Vglut2, and Vglut3 co-expression profiles in the central noradrenergic system and the combined measurements of breathing and oxygen consumption are two major strengths of this study. Observations from these experiments provide previously undescribed insights into (1) expression patterns for subtypes of the vesicular glutamate transporter protein in the noradrenergic system and (2) the dispensable nature of Vglut2dependent glutamate signaling from noradrenergic neurons to breathing responses to physiologically relevant gas challenges in adult conscious mice.Weaknesses:Although the cellular expression profiles for the vesicular glutamate transporters are provided, the study does not document that glutamatergic-based signaling originating from noradrenergic neurons is evident at the cellular level under normal, hypoxic, and/or hypercapnic conditions. The authors effectively recognize this issue and appropriately discuss their findings in this context.

We thank the reviewer for the positive evaluation of our work.

**Reviewer #2 (Public Review):**
The authors characterized the recombinase-based cumulative fate maps for vesicular glutamate transporters (Vglut1, Vglut2 and Vglut3) expression and compared those maps to their realtime expression profiles in central NA neurons by RNA in situ hybridization in adult mice. Authors have revealed a new and intriguing expression pattern for Vglut2, along with an entirely uncharted co-expression domain for Vglut3 within central noradrenergic neurons. Interestingly, and in contrast to previous studies, the authors demonstrated that glutamatergic signaling in central noradrenergic neurons does not exert any influence on breathing and metabolic control either under normoxic/normocapnic conditions or after chemoreflex stimulation. Also, they showed for the first-time the Vglut3-expressing NA population in C2/A2 nuclei. In addition, they were also able to demonstrate Vglut2 expression in anterior NA populations, such as LC neurons, by using more refined techniques, unlike previous studies.A major strength of the study is the use of a set of techniques to investigate the participation of NA-based glutamatergic signaling in breathing and metabolic control. The authors provided a full characterization of the recombinase-based cumulative fate maps for Vglut transporters. They performed real-time mRNA expression of Vglut transporters in central NA neurons of adult mice. Further, they evaluated the effect of knocking down Vglut2 expression in NA neurons using a DBH-Cre; Vglut2cKO mice on breathing and control in unanesthetized mice. Finally, they injected the AAV virus containing Cre-dependent Td tomato into LC of v-Glut2 Cre mice to verify the VGlut2 expression in LC-NA neurons. A very positive aspect of the article is that the authors combined ventilation with metabolic measurements. This integration holdsparticular significance, especially when delving into the exploration of respiratory chemosensitivity. Furthermore, the sample size of the experiments is excellent. Despite the clear strengths of the paper, some weaknesses exist. It is not clear in the manuscript if the experiments were performed in males and females and if the data were combined. I believe that the study would have benefited from a more comprehensive analysis exploring the sex specific differences. The reason I think this is particularly relevant is the developmental disorders mentioned by the authors, such as SIDS and Rett syndrome, which could potentially arise from disruptions in central noradrenergic (NA) function, exhibit varying degrees of sex predominance. Moreover, some of the noradrenergic cell groups are sexually dimorphic. For instance, female Wistar rats exhibit a larger LC size and more LC-NA neurons than male subjects (Pinos et al., 2001; Garcia-Falgueras et al., 2005). More recently, a detailed transcriptional profiling investigation has unveiled the identities of over 3,000 genes in the LC. This revelation has highlighted significant sexual dimorphisms, with more than 100 genes exhibiting differential expression within LC-NA neurons at the transcript level. Furthermore, this investigation has convincingly showcased that these distinct gene expression patterns have the capacity to elicit disparate behavioral responses between sexes (Mulvey et al., 2018).Therefore, the authors should compare the fate maps, Vglut transporters in males and females, at least considering LC-NA neurons. Even in the absence of identified sex differences, this information retains significant importance.An important point well raised by the authors is that although suggestive, these experiments do not definitively rule out that NA-Vglut2 based glutamatergic signaling has a role in breathing control. Subsequent experiments will be necessary to validate this hypothesis.An improvement could be made in terms of measuring body temperature. Opting for implanted sensors over rectal probes would circumvent the need to open the chamber, thereby preventing alterations in gas composition during respiratory measurements. Further, what happens to body temperature phenotype in these animals under different gas exposures? These data should be included in the Tables.Is it plausible that another neurotransmitter within NA neurons might be released in higher amounts in DBH-Cre; Vglut2 cKO mice to compensate for the deficiency in glutamate and prevent changes in ventilation?Continuing along the same line of inquiry is there a possibility that Vglut2 cKO from NA neurons not only eliminates glutamate release but also reduces NA release? A similar mechanism was previously found in VGLUT2 cKO from DA neurons in previous studies (Alsio et al., 2011; Fortin et al., 2012; Hnasko et al., 2010). Additionally, does glutamate play a role in the vesicular loading of NA? Therefore, could the lack of effect on breathing be explained by the lack of noradrenaline and not glutamate?

We thank the reviewer for the positive evaluation and further suggestions. Please see our response in “Author Response” to the previous version of Reviewer #2 (Public review).

**Reviewer #4 (Public Review):**
Summary:Although previous research suggested that noradrenergic glutamatergic signaling could influence respiratory control, the work performed by Chang and colleagues reveals that excitatory (specifically Vglut2) neurons is dynamically and widely expressed throughout the central noradrenergic system, but it is not significantly crucial to change baseline breathing as well the hypercapnia and hypoxia ventilatory responses. The central point that will make a significant change in the field is how NA-glutamate transmission may influence breathing control and the dysfunction of NA neurons in respiratory disorders.Strengths:There are several strengths such as the comprehensive analysis of Vglut1, Vglut2, and Vglut3 expression in the central noradrenergic system and the combined measurements of breathing parameters in conscious unrestrained mice.Other considerations :These results strongly suggest that glutamate may not be necessary for modulating breathing under normal conditions or even when faced with high levels of carbon dioxide (hypercapnia) or low oxygen levels (hypoxia). This finding is unexpected, considering many studies have underscored glutamate's vital role in respiratory regulation, more so than catecholamines. This leads us to question the significance of catecholamines in controlling respiration. Moreover, if glutamate is not essential for this function, we need to explore its role in other physiological processes such as sympathetic nerve activity (SNA), thermoregulation, and sensory physiology.

We thank the reviewer for the positive evaluation and further suggestions. The potential role of noradrenergic-derived glutamate in other processes, which is beyond the scope of this study, should be addressed in the future.

**Recommendations for the authors:**

**Reviewer #2 (Recommendations For The Authors):**
All of my concerns were effectively resolved, leading me to accept the paper. However, I suggest that the authors consider investing in a more reliable system for measuring body temperature, as accurate measurements of this parameter are crucial for whole body plethysmography.

Thank you for the suggestion. The real-time measurement of body temperature is a goal in future studies.

**Reviewer #4 (Recommendations For The Authors):**
Because I am revising a revised version, I believe the authors have addressed most, if not all, the concerns raised by already 3 reviewers. In my understanding the authors achieved their aims and the results are totally supported by the conclusions. The impact of this work on the respiratory field is significant and is likely to advance the field. The methods and data utilized, which combine standard techniques with genetic tools, will be highly beneficial to the research community.In my understanding I still have one concern that if glutamate is not critical, then what is? Could we potentially disable the noradrenergic (NA) system while preserving glutamate functionality to determine if the NA system is indeed crucial for respiratory physiology? This approach might provide clearer insights into the mechanisms underlying respiratory control.

We agree that there remain several exciting questions about the respective roles of noradrenaline, glutamate, and other neuropeptides such as Neuropeptide Y (NPY) and galanin. We are currently devising strategies to address the respective and combinatorial roles for all these candidates in breathing control. Most simply, we can conditionally, mutagenized each of them in the central noradrenergic system in an acute manner using DBH-CreER mice to determine if any of them are critical to respiratory control with the advantage of minimizing developmental compensatory events.